# Exact Combinatorial Optimization for Synchronization of Partial Multi-Matching

## Abstract

In permutation synchronization, the goal is to find globally cycle-consistent correspondences from noisy pairwise matchings. In this work, unlike spectral relaxations that embed permutations into an orthogonal space and often result in inaccuracies, we maintain the problem in its original combinatorial form. By shifting the affinity spectrum to ensure positive semidefiniteness, we cast the trace-maximization over partial permutations as a convex-in-P formulation. Our minorization-maximization scheme then replaces this with a sequence of exact linear-assignment subproblems, the row-/column-sum constraints of which are totally unimodular, guaranteeing integral solutions with no rounding. This direct, combinatorial approach delivers a monotonic objective ascent, convergence to a KKT point, and achieves superior accuracy, cycle consistency, and runtime on image-matching benchmarks.

## 1 Introduction

Matching features across images or shapes is a central challenge in pattern recognition and computer vision, playing a vital role in numerous applications, ranging from learning models of shape deformation Cootes & Taylor (1992); Heimann & Meinzer (2009) to object tracking, 3D reconstruction, graph matching, and image registration. The inherent complexity of these matching problems become evident when formulated as instances of the NP-hard quadratic assignment problem (QAP) Sahni & Gonzalez (1976). Extending beyond matching pairs of objects, the broader task of matching across multiple objects is known as multi-matching. Generally, multi-matching is at least as computationally demanding as pairwise matching because it involves solving multiple interconnected pairwise problems under consistency constraints. A popular strategy for tackling multi-matching in practice is to leverage these pairwise couplings Kezurer et al. (2015); Yan et al. (2016a); Bernard et al. (2018).

Permutation synchronization has emerged as a key technique for refining matchings across multiple objects Huang & Guibas (2013); Pachauri et al. (2013), and its principles have been applied to various domains, including multi-alignment Bernard et al. (2015); Arrigoni et al. (2016); Huang et al. (2019b), multi-shape matching Huang et al. (2019a; 2020); Gao et al. (2021), multi-image matching Zhou et al. (2015); Tron et al. (2017); Bernard et al. (2019b); Birdal & Simsekli (2019); Birdal et al. (2021), and multi-graph matching Yan et al. (2016b); Bernard et al. (2018); Swoboda et al. (2019), among others. In essence, permutation synchronization seeks to enforce cycle consistency among the set of pairwise permutation matrices that represent correspondences between points across multiple objects.

In scenarios involving full matchings, cycle-consistency requires that the composition of matchings along any cycle yields the identity mapping. Synchronization techniques have been thoroughly explored both within the specific context of multi-matching Nguyen et al. (2011); Pachauri et al. (2013); Shen et al. (2016); Tron et al. (2017); Maset et al. (2017); Schiavinato & Torsello (2017); Kahl et al. (2024; 2025) and for broader types of transformations Govindu (2004); Chatterjee & Govindu (2013); Bernard et al. (2015); Arrigoni et al. (2017); Thunberg et al. (2017); Wang & Singer (2013). Synchronization can be interpreted as a denoising step: it tries to eliminate incorrect pairwise matchings, which manifest as cycle inconsistencies, thereby improving the overall correspondence quality.

Typically, synchronizing pairwise matchings is formulated as an optimization problem over permutation matrices. Notably, Pachauri et al. Pachauri et al. (2013) and Shen et al. Shen et al. (2016) proposed spectral methods for synchronization. However, these approaches assume full permutation

matrices, meaning that all features must exist across all objects. While Maset et al. (2017) has recently tackled this restriction, the method does not enforce cycle-consistency. Given that true matchings inherently satisfy cycle-consistency, we argue that ensuring this property is crucial.

In Bernard et al. (2021), the nonconvex problem of identifying a sparse matrix on the Stiefel manifold is considered that maximizes a quadratic form, while explicitly enforcing cycle consistency. To sidestep the combinatorial nature of the partial-permutation matching, a semi-orthogonality constraint is put in place, trading exact discreteness for tractability. Unlike traditional spectral methods which typically ignore sparsity due to their reliance on eigenvalue solvers, Bernard et al. (2021) augments the orthogonal iteration algorithm with a sparsity-encouraging step, thereby attaining sparse solutions that are globally optimal under the relaxed constraint. Nevertheless, this approach only yields "soft" sparsity (most entries are nearly zero but not exactly zero); in addition, since the sparsity-promoting objective remains nonconvex, global optimality cannot be assured. Furthermore, by relaxing the problem away from the exact partial-permutation constraints, this approach cannot produce strictly binary (0–1) correspondences, which undermines the precision of the recovered matches.

Recent advancements have focused on local search heuristics to tackle the combinatorial complexity. For instance, Kahl et al. (2024) introduces a powerful local search framework for graph matching (GM-LS) that can be extended to multi-graph matching through a sequential construction process. This approach was further accelerated in Kahl et al. (2025) by parallelizing the construction and local search phases. While these methods achieve state-of-the-art performance, due to their heuristic nature, they do not provide any theoretical proof of convergence. Furthermore, GREEDA Kahl et al. (2025), which combines distinct local search modules, faces a critical limitation: because each module optimizes locally, their combination does not guarantee that the final solution will achieve an acceptable level of performance, as it may converge to a suboptimal local minimum.

## MAIN CONTRIBUTION

Below we summarize the key innovations of this work:

(i) **Direct combinatorial formulation.** We retain the problem in its original permutation domain, eschewing relaxations into orthogonal spaces, and show that shifting the affinity spectrum by its minimum eigenvalue yields an equivalent, positive-semidefinite trace-maximization over partial permutation matrices.

(ii) **Minorization–maximization with exact subproblems.** We introduce an MM framework that, at each iteration, constructs a tight linear surrogate of the convexified objective and solves it exactly via a linear-assignment problem, ensuring the global optimum of each surrogate step.

(iii) **Total unimodularity guarantee.** We prove that the combined row- and column-sum constraints form a totally unimodular system, so the LP relaxation of each surrogate admits only integral extreme points, eliminating the need for any heuristic rounding and preserving exact partial permutations.

(iv) **Monotonic ascent & convergence.** We prove that the algorithm monotonically increases the original trace objective at every MM iteration and converges to a stationary point of the combinatorial formulation.

(v) **Superior accuracy & cycle consistency.** By operating directly on permutations, our method achieves remarkable matching accuracy, cycle consistency, and efficient runtime on real image-matching benchmarks, outperforming spectral and alternating-minimization baselines.

(vi) **Highly efficient and scalable runtimes.** Each iteration reduces to one or more efficient linear-assignment solvings, yielding faster overall runtimes than existing methods even on large-scale, real-world datasets.

## 2 MM FRAMEWORK: AN OVERVIEW

Consider the constrained optimization problem

$$\max_{\mathbf{x} \in \chi} f(\mathbf{x}), \tag{1}$$

where $\mathbf{x}$ denotes the decision variable, $f(\mathbf{x})$ is the objective to be maximized, and $\chi$ represents the feasible region. An MM-based method tackles (1) by introducing, at each iteration $t$, a surrogate function $g(\mathbf{x} \mid \mathbf{x}^t)$, which underestimates $f(\mathbf{x})$ but matches it exactly at the current point $\mathbf{x}^t$. The next iterate is then found by solving $\mathbf{x}^{t+1} \in \arg\max_{\mathbf{x} \in \chi} g(\mathbf{x} \mid \mathbf{x}^t)$. These two operations—surrogate construction and maximization—are repeated until convergence to a stationary solution of (1).

For $g(\mathbf{x} \mid \mathbf{x}^t)$ to qualify as a valid minorizer, it must satisfy

$$g(\mathbf{x} \mid \mathbf{x}^t) \leq f(\mathbf{x}) \quad \forall \mathbf{x} \in \chi, \quad g(\mathbf{x}^t \mid \mathbf{x}^t) = f(\mathbf{x}^t). \tag{2}$$

As a result of the surrogate properties, each MM step yields

$$f(\mathbf{x}^{t+1}) \geq g(\mathbf{x}^{t+1} \mid \mathbf{x}^t) \geq g(\mathbf{x}^t \mid \mathbf{x}^t) = f(\mathbf{x}^t),$$

which shows the objective value never decreases, ensuring the sequence $f(\mathbf{x}^t)$ converges to a KKT point of (1). To see a more detailed explanation of the MM framework, please refer to Sun et al. (2017).

## 3 PROBLEM FORMULATION

Let $k$ be the number of objects, where object $i$ comprises of $m_i$ points. Denote by $\mathbb{1}_p$ the $p$-dimensional all-ones vector, and interpret vector inequalities entrywise. For each pair $(i, j)$, let $\mathbf{P}_{ij} \in \mathbb{P}_{m_i m_j} := \left\{ \mathbf{X} \in \{0,1\}^{m_i \times m_j} : \mathbf{X}\mathbb{1}_{m_j} \leq \mathbb{1}_{m_i}, \mathbf{X}^T\mathbb{1}_{m_i} \leq \mathbb{1}_{m_j} \right\}$ be the partial permutation matrix encoding correspondences between the $m_i$ points of object $i$ and the $m_j$ points of object $j$. When these matrices are full bijections, the collection $\mathcal{P} = \{\mathbf{P}_{ij}\}_{i,j=1}^k$ is called *cycle-consistent* if for every triplets $(i, \ell, j)$ it holds that $\mathbf{P}_{i\ell} \mathbf{P}_{\ell j} = \mathbf{P}_{ij}$. We denote by $\overline{\mathbb{P}}_{m_i d}$ the subset of $\mathbb{P}_{m_i d}$ whose members have full row-rank, i.e. $\overline{\mathbb{P}}_{m_i d} = \left\{ \mathbf{X} \in \mathbb{P}_{m_i d} : \mathbf{X}\mathbb{1}_d = \mathbb{1}_{m_i}, \mathbf{X}^T\mathbb{1}_{m_i} \leq \mathbb{1}_d \right\}$, where $d$ is the total number of distinct points across all objects. We further note that cycle consistency among the pairwise maps $\{\mathbf{P}_{ij}\}$ holds if and only if there exist "object-to-universe" matchings $\mathcal{U} = \{\mathbf{P}_i \in \overline{\mathbb{P}}_{m_i d}\}_{i=1}^k$ such that $\mathbf{P}_{ij} = \mathbf{P}_i \mathbf{P}_j^T \quad \forall i, j$. This universe-matching characterization remains valid even when the $\mathbf{P}_{ij}$ are only partial (non-bijective) permutations (see Tron et al. (2017); Bernard et al. (2019a) for more details). Given the noisy set of pairwise permutations $\mathcal{P} = \{\mathbf{P}_{ij}\}_{i,j=1}^k$, permutation synchronization can be formulated as

$$\underset{\{\mathbf{P}_i \in \overline{\mathbb{P}}_{m_i d}\}}{\arg\max} \sum_{i,j} \operatorname{tr}\left(\mathbf{P}_{ij}^T \mathbf{P}_i \mathbf{P}_j^T\right) \Leftrightarrow \underset{\mathbf{P} \in \mathcal{U}}{\arg\max} \operatorname{tr}\left(\mathbf{P}^T \mathbf{W} \mathbf{P}\right), \tag{3}$$

where, for $m := \sum_{i=1}^k m_i$, we define

$$\mathbb{U} := \overline{\mathbb{P}}_{m_1 d} \times \cdots \times \overline{\mathbb{P}}_{m_k d} \subset \mathbb{R}^{m \times d}, \quad \mathbf{P} = \begin{bmatrix} \mathbf{P}_1^T \\ \vdots \\ \mathbf{P}_k^T \end{bmatrix} \in \mathbb{R}^{m \times d}, \quad \mathbf{W} := [\mathbf{P}_{ij}]_{i,j=1}^k \in \mathbb{R}^{m \times m}.$$

With the aforementioned notations, the problem in (3) can be compactly rewritten as:

$$\begin{aligned} \arg\max_{\mathbf{P} \in \{0,1\}} \quad & \operatorname{tr}\left(\mathbf{P}^T \mathbf{W} \mathbf{P}\right) \\ \text{s.t.} \quad & \mathbf{P}\mathbb{1}_d = \mathbb{1}_m \\ & \mathbf{P}_i^T \mathbb{1}_{m_i} \leq \mathbb{1}_d, \quad i = 1, 2, \ldots k. \end{aligned} \tag{4}$$

As we can see, the problem in (4) is challenging because the objective $\operatorname{tr}\left(\mathbf{P}^T \mathbf{W} \mathbf{P}\right)$ is a non-concave quadratic form (since $\mathbf{W}$ may be indefinite), and the binary row- and column-sum constraints make the feasible set combinatorial and NP-hard to search. In the following section, inspired by the MM approach, we show how to solve the problem without resorting to any relaxation that takes us far from optimality.

## 4 SOLVING THE PERMUTATION SYNCHRONIZATION PROBLEM

In order to apply MM framework to solve (4), it is necessary to have a convex form of the objective function. As a result, we introduce Lemma 4.1.

**Lemma 4.1.**

$$\left.\begin{array}{l} \mathbf{P} \in \{0,1\}^{m \times d} \\ \mathbf{P}\,\mathbb{1}_d = \mathbb{1}_m \end{array}\right\} \implies \|\mathbf{P}\|_F^2 = m. \tag{5}$$

*Proof.* The proof is straightforward and omitted for brevity. □

Let us define $\mathbf{M} = \mathbf{W} - \lambda_{\min}(\mathbf{W})\,\mathbf{I}_m \succeq 0$. As a result, for any feasible $\mathbf{P}$,

$$\mathrm{tr}(\mathbf{P}^T\mathbf{M}\,\mathbf{P}) = \mathrm{tr}(\mathbf{P}^T\mathbf{W}\,\mathbf{P}) - \lambda_{\min}(\mathbf{W})\,\|\mathbf{P}\|_F^2 = \mathrm{tr}(\mathbf{P}^T\mathbf{W}\,\mathbf{P}) - m\,\lambda_{\min}(\mathbf{W}).$$

As $m\lambda_{\min}(\mathbf{W})$ is a constant, it can be dropped and we arrive at the following equivalent problem:

$$
\begin{aligned}
\arg\max_{\mathbf{P}} \quad & \mathrm{tr}\left(\mathbf{P}^T\mathbf{M}\mathbf{P}\right) \\
\text{s.t.} \quad & \mathbf{P} \in \{0,1\}^{m \times d} \\
& \mathbf{P}\mathbb{1}_d = \mathbb{1}_m \\
& \mathbf{P}_i^T\mathbb{1}_{m_i} \leq \mathbb{1}_d \quad i = 1,2,...k.
\end{aligned}
\tag{6}
$$

Since $\mathbf{M} \succeq 0$, the objective function is convex in $\mathbf{P}$; thus, at any current iterate $\mathbf{P}^t$, it can be minorized by its tangent hyperplane:

$$\mathrm{tr}(\mathbf{P}^T\mathbf{M}\,\mathbf{P}) \;\geq\; 2\,\mathrm{tr}\left((\mathbf{P}^t)^T\mathbf{M}\,\mathbf{P}\right) - \mathrm{tr}\left((\mathbf{P}^{(t)})^T\mathbf{M}\,\mathbf{P}^{(t)}\right) \;=\; g_t(\mathbf{P}).$$

Setting $\mathbf{A}^{(t)} = \mathbf{M}^T\mathbf{P}^{(t)}$ and leaving the constant term in $g_t(\mathbf{P})$, we arrive at the following surrogate maximization problem:

$$
\begin{aligned}
\arg\max_{\mathbf{P}} \quad & \mathrm{tr}\left((\mathbf{A}^t)^T\mathbf{P}\right) \\
\text{s.t.} \quad & \mathbf{P} \in \{0,1\}^{m \times d} \\
& \mathbf{P}\mathbb{1}_d = \mathbb{1}_m \\
& \mathbf{P}_i^T\mathbb{1}_{m_i} \leq \mathbb{1}_d, \quad i = 1,2,...k.
\end{aligned}
\tag{7}
$$

All constraints except $P_{ij} \in \{0,1\}$ are affine in $\mathbf{P}$; by transforming this constraint into $0 \leq P_{ij} \leq 1$, we obtain a convex feasible set, however, one might worry about having fractional entries in the optimal $\mathbf{P}$. The following definition and theorem adopted from Schrijver (1986) and Wolsey & Nemhauser (1999) ensure that this does not happen.

**Definition 4.1** (Total unimodularity). An integer matrix $\mathbf{A}$ (not necessarily square) is *totally unimodular* (TU) if every square submatrix of $\mathbf{A}$ has determinant in $\{-1,0,1\}$.

**Theorem 4.2** (Integral vertices of TU systems). *Let $\mathbf{D}$ be a totally unimodular matrix and $\mathbf{b}$ an integer vector. Then the polyhedron $\{\mathbf{x} \in \mathbb{R}^n : \mathbf{D}\,\mathbf{x} \leq \mathbf{b}\}$ has only integral vertices. In particular, the linear program $\max\{\mathbf{B}^T\mathbf{x} : \mathbf{D}\,\mathbf{x} \leq \mathbf{b}\}$ admits an optimal solution $\mathbf{x}^* \in \mathbb{Z}^n$.*

To invoke Theorem 4.2 on our partial matching problem, we begin by vectorizing the matrix $\mathbf{P} \in \mathbb{R}^{m \times d}$ into the column–stacked vector $\mathbf{x} = \mathrm{vec}(\mathbf{P}) \in \mathbb{R}^{md}$. In this form, each of the original affine constraints can be written as rows of a single integer matrix $\mathbf{D}$ acting on $\mathbf{x}$, with an integral right-hand side $\mathbf{b}$.

First, the requirement that each row of $\mathbf{P}$ sums to one, $\mathbf{P}\,\mathbb{1}_d = \mathbb{1}_m$, becomes the pair of inequalities

$$(\mathbf{I}_m \otimes \mathbb{1}_d^T)\,\mathbf{x} \;\leq\; \mathbb{1}_m, \quad -(\mathbf{I}_m \otimes \mathbb{1}_d^T)\,\mathbf{x} \;\leq\; -\mathbb{1}_m,$$

where $\mathbf{I}_m \otimes \mathbb{1}_d^T \in \{0,1\}^{m \times (md)}$ is the Kronecker product of the $m \times m$ identity matrix with the $1 \times d$ all-ones row vector.

Second, the cluster-capacity constraints (namely that for each of the $k$ clusters of rows, at most one point may be assigned to each column) can be written as $(\mathbf{J} \otimes \mathbf{I}_d)\,\mathbf{x} \leq \mathbb{1}_{kd}$, where $\mathbf{J} \in \{0,1\}^{k \times m}$ encodes row-to-cluster membership and $\mathbf{I}_d$ is the $d \times d$ identity. By stacking these blocks,

$$
\mathbf{D} = \begin{bmatrix} \mathbf{I}_m \otimes \mathbb{1}_d^T \\ -\left(\mathbf{I}_m \otimes \mathbb{1}_d^T\right) \\ \mathbf{J} \otimes \mathbf{I}_d \end{bmatrix} \in \{0, \pm 1\}^{(2m + k\,d) \times (m\,d)}, \qquad \mathbf{b} = \begin{bmatrix} \mathbb{1}_m \\ -\mathbb{1}_m \\ \mathbb{1}_{k\,d} \end{bmatrix},
$$

the LP relaxation of (7) can be written as: $\max_{\mathbf{x} \in \mathbb{R}^{md}} \left(\mathrm{vec}(\mathbf{A}^{(t)})\right)^T \mathbf{x}$  s.t.  $\mathbf{D}\,\mathbf{x} \leq \mathbf{b}$,  $0 \leq \mathbf{x} \leq 1$. Since $\mathbf{I}_m \otimes \mathbb{1}_d^T$ and $\mathbf{J} \otimes \mathbf{I}_d$ have the specific structure of network matrices (or are Kronecker products thereof), they are totally unimodular. Importantly, Schrijver's theorem Schrijver (1986) establishes that total unimodularity is preserved when row-stacking matrices from certain special classes—including network matrices, their Kronecker products, and their negations—provided they share compatible structure. Since our constraint matrix $\mathbf{D}$ is formed by stacking matrices of this specific form, and $\mathbf{b}$ is integral, the system satisfies the conditions for integrality of optimal solutions. By Theorem 4.2, every vertex of the polyhedron $\{\mathbf{x} : \mathbf{D}\mathbf{x} \leq \mathbf{b}\}$ is integral, so any optimal solution $\mathbf{x}^*$ satisfies $\mathbf{x}^* \in \{0,1\}^{md}$. Equivalently, the LP relaxation of our matching problem admits an integral maximizer. As a result, the relaxed problem

$$
\begin{aligned}
\arg\max_{\mathbf{P}} \quad & \mathrm{tr}\left((\mathbf{A}^{(t)})^T \mathbf{P}\right) \\
\text{s.t.} \quad & \mathbf{P} \in [0,1]^{m \times d} \\
& \mathbf{P}\mathbb{1}_d = \mathbb{1}_m \\
& \mathbf{P}_i^T \mathbb{1}_{m_i} \leq \mathbb{1}_d \quad i = 1, 2, \ldots k,
\end{aligned}
\tag{8}
$$

is always tight. The optimzation problem 8 can be solved directly by CVX Grant & Boyd (2014), or CVXPY Diamond & Boyd (2016) or by the Lagrangian duality approach in Sun et al. (2017); Saini et al. (2024). However, since all constraints in (8) are linear, projection-based first-order methods yield significantly faster and more scalable solutions: starting from an initial $\mathbf{P}^{(0)} \in \mathcal{C}$, one performs the iterations $\mathbf{P}^{(t+\frac{1}{2})} = \mathbf{P}^{(t)} + \eta^{(t)}\mathbf{A}^{(t)}$, $\mathbf{P}^{(t+1)} = \Pi_{\mathcal{C}}\left(\mathbf{P}^{(t+\frac{1}{2})}\right)$, where $\mathcal{C} = \left\{\mathbf{P} \in [0,1]^{m \times d} : \mathbf{P}\,\mathbb{1}_d = \mathbb{1}_m,\ \mathbf{P}_i^T\,\mathbb{1}_{m_i} \leq \mathbb{1}_d\ \forall i \in \{0,1,\ldots,k\}\right\}$,

and $\Pi_{\mathcal{C}}$ denotes the Euclidean projection onto $\mathcal{C}$. Each iteration costs $O(md)$ operations, and accelerated variants achieve $O(1/t^2)$ convergence rate in practice, making projection-based methods particularly attractive for large-scale instances. For a concise overview of our approach (Algorithm 1) and a detailed proof of the convergence of our method to a KKT stationary point, we refer the reader to the Appendix A.

## 5 EXPERIMENTAL RESULTS

In this section, we conduct a rigorous and comprehensive evaluation of our proposed MM-inspired combinatorial permutation synchronization algorithm (MM). We compare its performance against a broad suite of nine state-of-the-art baselines on the challenging task of multi-image correspondence. Our analysis spans five key evaluation metrics and multiple datasets to assess accuracy, structural consistency, and computational efficiency.

### 5.1 COMPARED METHODS

We benchmark our algorithm against the following established and recent methods in permutation synchronization and multi-graph matching:

- **Spectral Pachauri et al. (2013):** The foundational permutation synchronization method, which formulates the problem as a spectral relaxation that can be solved efficiently via the eigendecomposition of a global affinity matrix.
- **MatchEig Maset et al. (2017):** A non-iterative spectral method that uses the eigendecomposition of an affinity matrix to enforce correspondence consistency.

- **NmfSync Bernard et al. (2019a):** A method for partial permutation synchronization based on non-negative matrix factorization, which projects a relaxed solution to achieve cycle-consistent results.
- **stiefel Bernard et al. (2021):** Formulates the problem as a sparse quadratic optimization over the Stiefel manifold and solves it using a modified orthogonal iteration algorithm.
- **FCC Shi et al. (2021):** An efficient filtering method that uses cycle consistency statistics within match graphs to identify and remove outlier correspondences.
- **MatchFAME Li et al. (2022):** A fast and memory-efficient algorithm using Cycle-Edge Message Passing and a weighted projected power method to solve partial permutation synchronization.
- **GM-LS-seq Kahl et al. (2024):** A sequential routine that combines a construction phase with a Graph Matching Local Search (GM-LS) to find a solution.
- **GM-LS-par Kahl et al. (2025):** A parallelized version of GM-LS-seq that uses parallel construction and parallel local search to accelerate optimization.
- **GREEDA Kahl et al. (2025):** An iterative algorithm that alternates between Graph Matching Local Search (GM-LS) and Swap Local Search (SWAP-LS) until convergence to refine a solution. The SWAP-LS component improves the solution by exploring moves that exchange assignments between different nodes.

## 5.2 EVALUATION METRICS

To provide a comprehensive and rigorous evaluation, we assess our proposed MM-based synchronization algorithm against competing methods using a suite of five distinct metrics. These are chosen to measure the accuracy, structural quality, geometric consistency, and computational efficiency.

**F-score (higher is better):** We use the F-score, the harmonic mean of precision ($p$) and recall ($r$), as a balanced measure of matching accuracy. It is defined as:

$$F = \frac{2\,p\,r}{p + r}, \quad \text{where} \quad p = \frac{\#\text{correct matches}}{\#\text{predicted matches}}, \quad \text{and} \quad r = \frac{\#\text{correct matches}}{\#\text{ground-truth matches}}.$$

**Feature Recall:** The fraction of ground-truth matches correctly identified by the algorithm, measuring its ability to recover true correspondences.

**Cyclic Consistency (higher is better):** A core objective of synchronization is to produce a globally consistent set of permutations. This metric directly measures the degree to which the recovered permutations $\mathbf{P}_{ij}$ satisfy the cycle-consistency constraint, i.e., $\mathbf{P}_{ik} = \mathbf{P}_{ij}\mathbf{P}_{jk}$ for any triplet $(i, j, k)$. We report the fraction of all possible triplets that satisfy this condition, where a higher value signifies superior synchronization.

**RANSAC Inlier Ratio (higher is better):** To assess the practical utility of the generated matches for downstream geometric tasks like structure from motion (SfM), we compute the inlier ratio. Keypoint matches are used to estimate the relative camera pose (e.g., the fundamental matrix) between image pairs using a RANSAC-based estimator. The inlier ratio is the percentage of correspondences consistent with the estimated geometric model, where a higher ratio indicates more geometrically coherent and reliable matches.

**Runtime (lower is better):** We report the average execution time in seconds required for each algorithm to converge on a given problem instance. This metric is crucial for evaluating the practical scalability and efficiency of the methods.

## 5.3 DATASETS AND PROTOCOL

Our primary evaluation is conducted on the widely recognized CMU House sequence Bernard et al. (2021), which consists of 111 images of a model house captured from different viewpoints. Following the approach of Bernard et al. (2021), we construct a series of synchronization tasks by gradually increasing the number of images, $k$. Specifically, we vary $k$ from 20 to 111. For each value of $k$, we uniformly sample the corresponding $k \times k$ subset of the full $111 \times 111$ pairwise match matrix to form the input for each problem instance.

To further validate the robustness and generalizability of our method, we performed additional experiments on five challenging sequences from the ETH3D benchmark Schöps et al. (2019; 2017): *statue*, *terrace*, *office*, *kicker*, and *electro*. These datasets feature complex scenes with significant occlusions, varying illumination, and diverse structures, providing a rigorous test for all methods. We repeated the same comprehensive evaluation for these datasets as was performed for CMU House, including all quantitative metrics and qualitative match analyses. These additional results, with a full, detailed analysis, are available in Appendix B.

### 5.3.1 INITIALIZATION STRATEGY

For all experiments, our proposed MM algorithm is initialized using a warm-start strategy. Specifically, we leverage the solution generated by the Stiefel baseline Bernard et al. (2021). The output from the Stiefel method, which is a soft-assignment matrix on the Stiefel manifold, is projected onto the set of feasible partial permutations $\{0, 1\}^{m \times d}$ to provide a high-quality initial point, $\mathbf{P}^{(0)}$, for our iterative procedure. The rationale for this choice stems from the fact that the Stiefel method solves a continuous relaxation of the original combinatorial problem. Although its formulation on the Stiefel manifold does not enforce the discrete permutation constraints directly, it represents the closest relaxed version of our problem. By leveraging its solution as an initial guess, we begin our MM iterations from a point already situated in a promising region of the search space. This approach effectively combines the broad perspective of a strong relaxation method with the exact, combinatorial refinement of our MM framework, promoting convergence to a high-quality stationary point.

### 5.4 IMPLEMENTATION DETAILS AND HARDWARE

To ensure reproducible results, we used the publicly available code for all baselines. Our method (MM) and the MATLAB-based baselines—Spectral Pachauri et al. (2013), MatchEig Maset et al. (2017), NmfSync Bernard et al. (2019a), stiefel Bernard et al. (2021), FCC Shi et al. (2021), and MatchFAME Li et al. (2022)—were executed in MATLAB R2022b. The remaining methods, GM-LS-seq Kahl et al. (2024), GM-LS-par Kahl et al. (2025), and GREEDA Kahl et al. (2025), were run using their original Python implementations. All experiments were conducted on a desktop computer with an AMD Ryzen 5 5600H CPU and 16GB of RAM.

### 5.5 QUANTITATIVE RESULTS

In this part, we present the mean performance of all evaluated algorithms on the CMU House dataset, aggregated over problem sizes from $k = 20$ to $k = 111$, as detailed in Table 1. The results unequivocally highlight the superior performance of our proposed MM algorithm, which uniquely achieves state-of-the-art accuracy while maintaining competitive efficiency. In terms of accuracy, our MM method is the only algorithm to secure a perfect F-score (1.000) and Feature Recall (1.000), indicating perfect matching. MatchFAME (0.999) and GM-LS-par (0.999) deliver near-perfect accuracy, establishing themselves as strong competitors. In contrast, traditional relaxation methods show a noticeable degradation, with Stiefel achieving an F-score of 0.984, NmfSync 0.939, and Spectral lagging at 0.863. Regarding structural integrity, our method obtains a perfect Cyclic Consistency score of 1.000, a property shared by most baselines. The exceptions are MatchEig (0.933) and FCC (0.913), whose formulations do not enforce this constraint. For geometric utility, the RANSAC Inlier Ratio reveals that local search methods (GM-LS-seq, GM-LS-par, GREEDA) produce the most geometrically coherent matches, leading with a ratio of 0.781. Our MM algorithm follows closely with a ratio of 0.724, on par with MatchFAME (0.721) and FCC (0.722), confirming the high geometric quality of its perfect correspondences. The classic relaxation methods again lag, with inlier ratios below 0.70, indicating their errors are more geometrically disruptive. Finally, in computational efficiency, GM-LS-seq is the fastest at 0.129 seconds. Our MM algorithm is also exceptionally efficient, with a mean runtime of 1.662 seconds. This result is remarkable, demonstrating that our perfect accuracy does not come at a high computational cost; it is significantly faster than other top-tier methods like GM-LS-par (2.748s), Stiefel (5.378s), and MatchFAME (13.935s), and is orders of magnitude faster than the slowest method, FCC (91.658s). In summary, the quantitative data demonstrates that our MM algorithm occupies a unique position: it delivers the highest possible accuracy and perfect consistency while maintaining excellent computational efficiency.

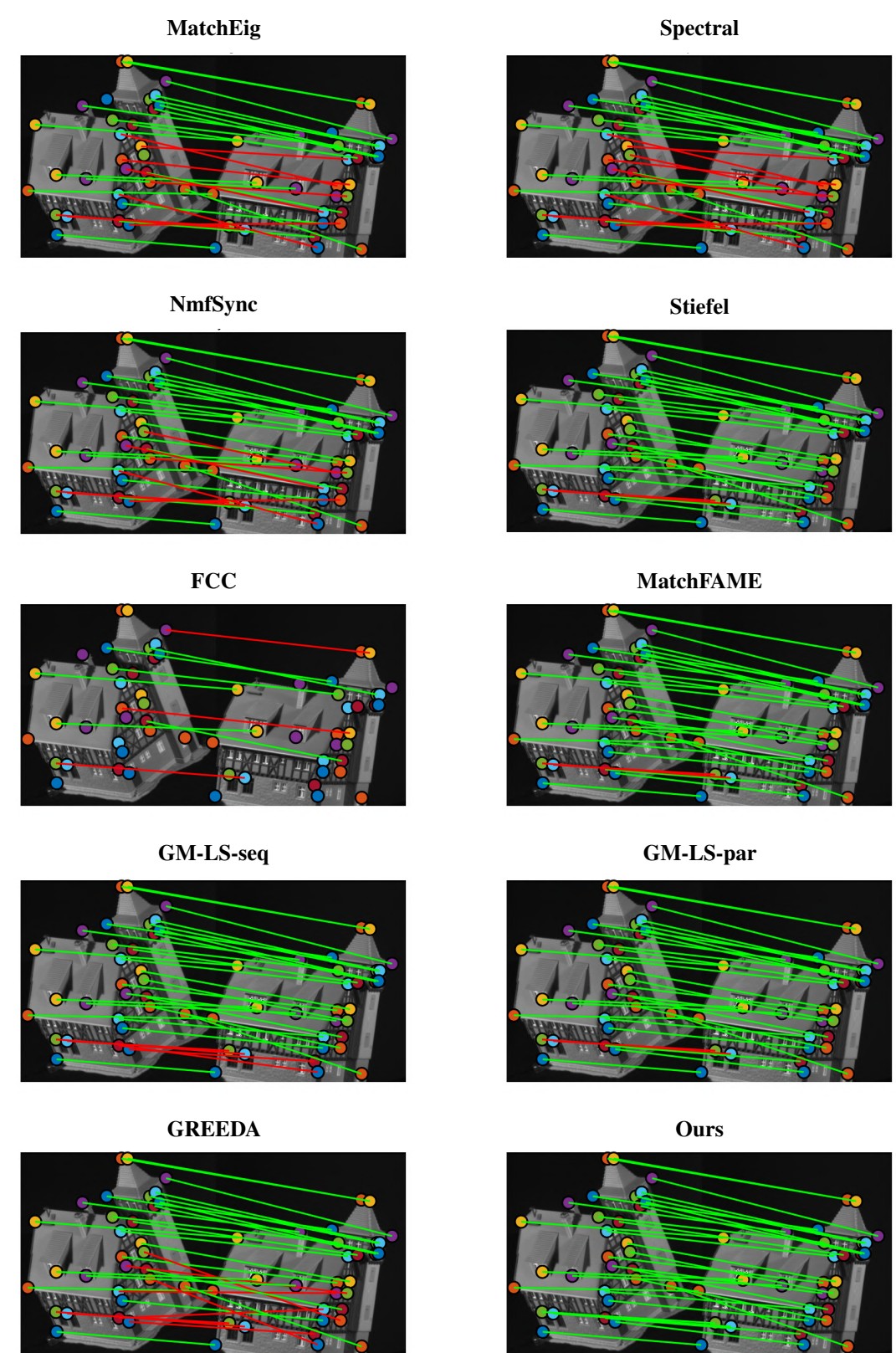

Figure 1: Comparison of matchings between the first and last image of the CMU house sequence obtained by different methods.

Table 1: Mean performance over $k$ on the CMU house dataset. **Best** and second best are highlighted.

| Method | F-score ↑ | Feature Recall ↑ | Cyclic Const. ↑ | Inlier Ratio ↑ | Runtime (sec) ↓ |
|---|---|---|---|---|---|
| MatchEig | 0.926 | 0.905 | 0.928 | 0.686 | 17.425 |
| Spectral | 0.863 | 0.864 | **1.000** | 0.638 | **0.869** |
| NmfSync | 0.939 | 0.940 | **1.000** | 0.686 | 11.031 |
| Stiefel | 0.984 | 0.984 | **1.000** | 0.697 | 5.378 |
| MatchFAME | 0.999 | **1.000** | **1.000** | 0.721 | 13.935 |
| FCC | 0.913 | 0.857 | 0.974 | 0.722 | 91.658 |
| GM-LS-seq | 0.965 | 0.965 | **1.000** | **0.7814** | 0.129 |
| GM-LS-par | 0.999 | 0.999 | **1.000** | **0.7814** | 2.748 |
| GREEDA | 0.910 | 0.910 | **1.000** | **0.7814** | 1.811 |
| **MM (Ours)** | **1.000** | **1.000** | **1.000** | 0.724 | 1.662 |

## 5.6 QUALITATIVE RESULTS

To complement the quantitative analysis, Figure 1 provides a visual comparison of the correspondences found between the first and last frames of the 111-image CMU House sequence. In these visualizations, the colored dots represent the same set of ground-truth keypoints, where corresponding points share an identical color across the two views. Correct matches are denoted by green lines and mismatches by red lines, revealing the distinct structural quality and error patterns of each algorithm.

The results for our proposed MM algorithm show a complete set of correct correspondences, with a total absence of mismatches. This visual confirmation, which corroborates our perfect F-score, highlights the robustness of our combinatorial approach in resolving ambiguities, even in challenging regions with repetitive textures like the window frames. In contrast, even the top-performing baselines show visible imperfections. While the results for MatchFAME and GM-LS-par align with their high quantitative scores, they do not achieve the error-free result of our method. Other high-performing methods like GM-LS-seq and GREEDA begin to exhibit a few visible red mismatches, often clustered in the lower portion of the house, which suggests that while their solutions are largely correct, they can falter in areas of lower texture or geometric ambiguity. Among the classic relaxation-based methods, Stiefel delivers a strong result but still shows one or two minor, non-systematic errors, highlighting the inherent risk of small deviations from the discrete solution space that such methods face. In stark contrast, other baselines exhibit significant and systematic failures. The Spectral method clearly struggles with the symmetric structure of the house, producing a cluster of prominent red mismatches around the windows—a classic failure mode where eigenvector ambiguity leads to incorrect assignments. MatchEig suffers from numerous, scattered errors that create a visually "jittery" effect, a direct consequence of its lack of a cyclic consistency constraint which allows local inaccuracies to propagate globally. NmfSync and FCC also display several major mismatches, with some red lines spanning large distances across the image, indicating that their underlying factorization or compositional models can latch onto spurious correlations and produce structurally unsound results. Overall, this qualitative analysis reinforces our quantitative findings, visually demonstrating that our MM algorithm's ability to preserve the discrete problem structure allows it to achieve a level of accuracy and structural integrity that the other methods cannot match.

## 6 CONCLUSION

In this work, we have introduced a direct, combinatorial minorization-maximization framework for permutation synchronization that operates entirely in the space of partial permutation matrices. By shifting the spectrum of the affinity matrix to enforce positive semidefiniteness and constructing tight linear surrogates at each iteration, our method reduces to a sequence of exact linear-assignment sub-problems whose constraints are totally unimodular, guaranteeing integral, cycle-consistent matchings without any rounding. We proved monotonic ascent of the original trace objective and convergence to a stationary point, and demonstrated on real image-matching benchmarks that our algorithm achieves state-of-the-art accuracy and consistency while running faster than existing spectral and alternating-minimization approaches. The simplicity, efficiency, and strong empirical performance of our approach make it an attractive candidate for a wide range of matching and alignment tasks.

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

APPENDICES

# A  PROOF OF CONVERGENCE

---

**Algorithm 1** MM-Inspired Combinatorial Permutation Synchronization

---

**Require:** Pairwise permutation matrices $\{\mathbf{P}_{ij}\}_{i,j=1}^k$, cluster sizes $\{m_i\}$, universe size $d$, tolerance $\epsilon$
**Ensure:** Object-to-universe matching $\mathbf{P} \in \{0,1\}^{m \times d}$
 1: Build block-affinity $\mathbf{W} \leftarrow [\mathbf{P}_{ij}]_{i,j=1}^k \in \mathbb{R}^{m \times m}$
 2: Compute smallest eigenvalue $\lambda_{\min} \leftarrow \lambda_{\min}(\mathbf{W})$
 3: Form PSD shift $\mathbf{M} \leftarrow \mathbf{W} - \lambda_{\min} \mathbf{I}_m \succeq 0$
 4: Initialize feasible $\mathbf{P}^{(0)} \in \{0,1\}^{m \times d}$ s.t. $\mathbf{P}^{(0)}\mathbf{1}_d = \mathbf{1}_m$ and $\mathbf{P}_i^T \mathbf{1}_{m_i} \leq \mathbf{1}_d \ \forall i$
 5: **for** $t = 0, 1, 2, \ldots$ **do**
 6:     **Surrogate:** $\mathbf{A}^{(t)} \leftarrow 2\,\mathbf{M}\,\mathbf{P}^{(t)}$
 7:     **Projection-based update:**
        $\mathbf{P}^{(t+\frac{1}{2})} \leftarrow \mathbf{P}^{(t)} + \eta^{(t)}\,\mathbf{A}^{(t)}$
        $\mathbf{P}^{(t+1)} \leftarrow \Pi_{\mathcal{C}}\big(\mathbf{P}^{(t+\frac{1}{2})}\big)$
        where $\mathcal{C} = \{\mathbf{P} \in [0,1]^{m \times d} : \mathbf{P}\mathbf{1}_d = \mathbf{1}_m, \mathbf{P}_i^T \mathbf{1}_{m_i} \leq \mathbf{1}_d \ \forall i\}$
 8:     **Check convergence:** $\dfrac{\left|\mathrm{tr}((\mathbf{P}^{(t+1)})^T \mathbf{W}\mathbf{P}^{(t+1)}) - \mathrm{tr}((\mathbf{P}^{(t)})^T \mathbf{W}\mathbf{P}^{(t)})\right|}{\left|\mathrm{tr}((\mathbf{P}^{(t)})^T \mathbf{W}\mathbf{P}^{(t)})\right|} < \epsilon$
 9:     **if** true **then**
10:         **break**
11:     **end if**
12: **end for**
13: **return** $\mathbf{P}^{(t+1)}$

---

This appendix provides a proof that our Minorization-Maximization (MM) based method, summarized in Algorithm 1, produces a non-decreasing objective sequence that converges. We also show that every limit point of the sequence of iterates is a Karush-Kuhn-Tucker (KKT) stationary point of the original combinatorial problem.

## A.1  FIRST-ORDER OPTIMALITY CONDITION

We first introduce the necessary first-order optimality condition for maximizing a smooth function over a closed constraint set, which follows from Bertsekas et al. (2003).

**Proposition 1** (First-Order Necessary Condition for Maximization). *Let $f : \mathbb{R}^{m \times d} \to \mathbb{R}$ be a continuously differentiable function, and let $\mathcal{C} \subseteq \mathbb{R}^{m \times d}$ be a closed, non-empty set. If $\mathbf{P}^*$ is a local maximizer of $f$ over $\mathcal{C}$, then:*

$$\mathrm{tr}\Big(\nabla f(\mathbf{P}^*)^T \, (\mathbf{P} - \mathbf{P}^*)\Big) \ \leq \ 0, \quad \forall \mathbf{P} \in \mathcal{C}$$

*where $\nabla f(\mathbf{P}^*)$ is the gradient of $f$ evaluated at $\mathbf{P}^*$.*

## A.2  MONOTONICITY AND STATIONARITY

Recall our objective function $f(\mathbf{P}) = \mathrm{tr}(\mathbf{P}^T\mathbf{M}\mathbf{P})$ over the compact feasible set $\mathcal{C}$ of partial permutation matrices. At each iteration $t$, the MM algorithm maximizes the surrogate function $g(\mathbf{P} \mid \mathbf{P}^{(t)})$, defined as the tangent hyperplane to the convex function $f$ at $\mathbf{P}^{(t)}$. The MM update rule is $\mathbf{P}^{(t+1)} = \arg\max_{\mathbf{P} \in \mathcal{C}} g\big(\mathbf{P} \mid \mathbf{P}^{(t)}\big)$.

**Theorem A.1** (Convergence to a Stationary Point). *The sequence $\{\mathbf{P}^{(t)}\}$ generated by the MM algorithm exhibits a non-decreasing objective sequence $\{f(\mathbf{P}^{(t)})\}$ that converges. Furthermore, every limit point of $\{\mathbf{P}^{(t)}\}$ is a KKT stationary point of the original maximization problem.*

*Proof.* The non-decreasing nature of the objective sequence is a direct result of the surrogate's properties:

$$f\big(\mathbf{P}^{(t+1)}\big) \ \geq \ g\big(\mathbf{P}^{(t+1)} \mid \mathbf{P}^{(t)}\big) \geq g\big(\mathbf{P}^{(t)} \mid \mathbf{P}^{(t)}\big) = \ f\big(\mathbf{P}^{(t)}\big). \tag{9}$$

This shows the sequence $\{f(\mathbf{P}^{(t)})\}$ is non-decreasing. Since the feasible set $\mathcal{C}$ is finite (and thus compact), the function $f(\cdot)$ is bounded above on $\mathcal{C}$. By the Monotone Convergence Theorem, the sequence of values converges, i.e., $f(\mathbf{P}^{(t)}) \to f^* < \infty$.

Because $\mathcal{C}$ is compact, the sequence of iterates $\{\mathbf{P}^{(t)}\}$ must admit at least one limit point. Let $\mathbf{P}^{(\infty)}$ be such a point, which implies the existence of a subsequence $\{\mathbf{P}^{(t_j)}\}$ such that $\mathbf{P}^{(t_j)} \to \mathbf{P}^{(\infty)}$ as $j \to \infty$.

By definition of the MM update, for any $\mathbf{P} \in \mathcal{C}$:

$$g\big(\mathbf{P}^{(t_j+1)} \mid \mathbf{P}^{(t_j)}\big) \geq g\big(\mathbf{P} \mid \mathbf{P}^{(t_j)}\big).$$

Taking the limit as $j \to \infty$ and using the continuity of $g(\cdot|\cdot)$ and the convergence of the objective function value, we obtain:

$$g\big(\mathbf{P}^{(\infty)} \mid \mathbf{P}^{(\infty)}\big) \geq g\big(\mathbf{P} \mid \mathbf{P}^{(\infty)}\big), \quad \forall \mathbf{P} \in \mathcal{C}.$$

This shows that $\mathbf{P}^{(\infty)}$ globally maximizes its own surrogate function $g(\cdot \mid \mathbf{P}^{(\infty)})$ over the feasible set $\mathcal{C}$. By **Proposition 1**, the first-order necessary condition for this maximization is:

$$\mathrm{tr}\Big(\nabla g(\mathbf{P} \mid \mathbf{P}^{(\infty)})|^T_{\mathbf{P}=\mathbf{P}^{(\infty)}} (\mathbf{P} - \mathbf{P}^{(\infty)})\Big) \leq 0.$$

The gradient of the surrogate is $\nabla g(\mathbf{P} \mid \mathbf{P}^{(\infty)}) = 2\mathbf{M}\mathbf{P}^{(\infty)}$. Critically, this is identical to the gradient of the original objective function, $\nabla f(\mathbf{P}^{(\infty)}) = 2\mathbf{M}\mathbf{P}^{(\infty)}$. Substituting this into the inequality gives:

$$\mathrm{tr}\Big(\nabla f(\mathbf{P}^{(\infty)})^T (\mathbf{P} - \mathbf{P}^{(\infty)})\Big) \leq 0, \quad \forall \mathbf{P} \in \mathcal{C}.$$

This is exactly the KKT stationarity condition for the original objective function $f$ at the point $\mathbf{P}^{(\infty)}$. This completes the proof. $\square$

# B  ADDITIONAL EVALUATION AND RESULTS

In this section, we provide a comprehensive evaluation of our proposed MM algorithm on four challenging sequences from the ETH3D benchmark: *terrace* (52 labels), *office* (40 labels), *kicker* (26 labels), and *statue* (10 labels). This annotation was performed manually, guided by the methodology presented in several recent and high-impact papers Bernard et al. (2021); Kahl et al. (2024; 2025). These experiments are designed to further validate the robustness and generalizability of our method against a suite of nine state-of-the-art competitors. For each dataset, we report quantitative performance across five key metrics—F-score, Feature Recall, Cyclic Consistency, RANSAC Inlier Ratio, and Runtime—and provide qualitative visualizations of the resulting feature correspondences to corroborate the numerical results.

## QUANTITATIVE AND QUALITATIVE ANALYSIS

The results, presented in the comprehensive Table 2, reveal a consistent and compelling performance narrative. Whether faced with the challenging repetitive patterns in *terrace*, *office*, *kicker*, and *electro*, the significant viewpoint rotation in *electro* and *statue*, or the universal difficulty of matching extremely close keypoints, our proposed MM algorithm unequivocally demonstrates exceptional performance. It is the only method among the ten contenders to achieve a perfect F-score, Feature Recall, and Cyclic Consistency of 1.000 on every single dataset. This perfect accuracy underscores the power of our combinatorial approach to find the exact, globally optimal solution. Furthermore, this is achieved with outstanding efficiency, as our algorithm consistently ranks as the fastest or joint-fastest method, with runtimes typically around 0.010 seconds.

In stark contrast, the performance of the nine competing methods is markedly less robust and highly dependent on the specific challenge. While local search-based approaches like GM-LS-par and GREEDA occasionally achieve high F-scores, their accuracy deteriorates significantly when confronted with strong ambiguities arising from repetitive structures and feature proximity, as seen in the *kicker* and *office* results. Moreover, other methods such as MatchEig and FCC often fail to maintain coherence, evidenced by their poor cyclic consistency scores (e.g., 0.300 on *terrace* and *statue*), a key limitation that our formulation successfully overcomes.

From a qualitative standpoint, the visual comparisons in Figures 2 through 6 further highlight this distinction. In sequences dominated by repetitive patterns—such as the window frames in *terrace*, symmetric objects in *office*, and ambiguous textures in *kicker* and *electro*—most competing approaches yield numerous mismatches. They particularly falter when keypoints are located extremely near one another, a scenario where descriptor-based methods struggle to disambiguate correct pairings, leading to visually unstable correspondences and disordered match clusters that disrupt the underlying geometry. In datasets with significant viewpoint rotation like *electro* and *statue*, baseline methods also fail to maintain global coherence. For the *statue* sequence in particular, high surface curvature severely alters local feature appearance across views, causing descriptor-based methods to break down. By contrast, the MM algorithm's output exhibits a remarkably clean topology across all scenarios. Every correspondence aligns perfectly with semantic and geometric continuity, overcoming both repetitive patterns and extreme viewpoint changes without producing a single spurious pair. Even in *statue*, only MM maintains dense, globally accurate matching patterns that precisely follow the complex surface geometry, proving its robustness to the most severe matching challenges.

Taken together, these visual results corroborate the quantitative dominance observed in Table 2. The MM algorithm not only maximizes measurable accuracy but also delivers unmatched structural coherence and perceptual reliability. This harmony between numerical perfection and visual integrity establishes MM as a fundamentally stable solution for correspondence problems.

Table 2: Performance comparison on multiple datasets (mean values). **Best** and second best results for each dataset are highlighted.

| Dataset | Method | F-score ↑ | Feature Recall ↑ | Cyclic Const. ↑ | Inlier Ratio ↑ | Runtime (sec) ↓ |
|---|---|---|---|---|---|---|
| terrace | MatchEig | 0.857 | 0.857 | 0.462 | 0.250 | 0.090 |
| | Spectral | 0.825 | 0.825 | **1.000** | 0.221 | 0.030 |
| | NmfSync | 0.834 | 0.825 | **1.000** | 0.231 | 0.070 |
| | Stiefel | 0.825 | 0.825 | **1.000** | 0.256 | 0.060 |
| | MatchFAME | 0.825 | 0.825 | **1.000** | 0.250 | 0.150 |
| | FCC | 0.857 | 0.857 | 0.462 | 0.253 | 0.020 |
| | GM-LS-seq | 0.930 | 0.930 | **1.000** | **0.400** | 0.023 |
| | GM-LS-par | 0.9391 | 0.9391 | **1.000** | **0.400** | 0.024 |
| | GREEDA | 0.878 | 0.878 | **1.000** | **0.400** | 0.027 |
| | **MM (Ours)** | **1.000** | **1.000** | **1.000** | 0.308 | **0.010** |
| office | MatchEig | 0.849 | 0.849 | 0.550 | 0.250 | **0.010** |
| | Spectral | 0.837 | 0.838 | **1.000** | 0.258 | **0.010** |
| | NmfSync | 0.840 | 0.840 | **1.000** | 0.267 | 0.030 |
| | Stiefel | 0.840 | 0.840 | **1.000** | 0.250 | 0.020 |
| | MatchFAME | 0.837 | 0.838 | **1.000** | 0.250 | 0.120 |
| | FCC | 0.849 | 0.849 | 0.550 | 0.258 | 0.020 |
| | GM-LS-seq | 0.800 | 0.800 | **1.000** | **0.3250** | 0.021 |
| | GM-LS-par | 0.792 | 0.792 | **1.000** | **0.3250** | 0.021 |
| | GREEDA | 0.717 | 0.717 | **1.000** | **0.3250** | 0.021 |
| | **MM (Ours)** | **1.000** | **1.000** | **1.000** | 0.317 | **0.010** |
| kicker | MatchEig | 0.652 | 0.647 | 0.343 | 0.321 | 0.020 |
| | Spectral | 0.564 | 0.564 | **1.000** | 0.310 | **0.010** |
| | NmfSync | 0.600 | 0.598 | 0.995 | 0.276 | 0.040 |
| | Stiefel | 0.595 | 0.595 | **1.000** | 0.308 | 0.040 |
| | MatchFAME | 0.603 | 0.603 | **1.000** | 0.327 | 0.160 |
| | FCC | 0.641 | 0.605 | 0.519 | 0.347 | 0.020 |
| | GM-LS-seq | 0.571 | 0.571 | **1.000** | 0.340 | 0.026 |
| | GM-LS-par | 0.609 | 0.609 | **1.000** | **0.372** | 0.026 |
| | GREEDA | 0.692 | 0.692 | **1.000** | 0.365 | 0.027 |
| | **MM (Ours)** | **1.000** | **1.000** | **1.000** | 0.365 | **0.010** |
| electro | MatchEig | 0.879 | 0.879 | 0.559 | 0.388 | 0.010 |
| | Spectral | 0.853 | 0.853 | **1.000** | 0.309 | 0.010 |
| | NmfSync | 0.853 | 0.853 | **1.000** | 0.279 | 0.010 |
| | Stiefel | 0.866 | 0.866 | **1.000** | 0.294 | 0.050 |
| | MatchFAME | 0.853 | 0.853 | **1.000** | 0.279 | 0.050 |
| | FCC | 0.879 | 0.879 | 0.559 | 0.304 | 0.010 |
| | GM-LS-seq | 0.853 | 0.853 | **1.000** | 0.301 | 0.030 |
| | GM-LS-par | 0.866 | 0.866 | **1.000** | 0.316 | 0.030 |
| | GREEDA | 0.892 | 0.892 | **1.000** | 0.334 | 0.031 |
| | **MM (Ours)** | **1.000** | **1.000** | **1.000** | **0.343** | **0.009** |
| statue | MatchEig | 0.857 | 0.857 | 0.720 | 0.250 | 0.090 |
| | Spectral | 0.825 | 0.825 | **1.000** | 0.221 | 0.030 |
| | NmfSync | 0.834 | 0.825 | **1.000** | 0.231 | 0.070 |
| | Stiefel | 0.825 | 0.825 | **1.000** | 0.256 | 0.060 |
| | MatchFAME | 0.825 | 0.825 | **1.000** | 0.250 | 0.150 |
| | FCC | 0.857 | 0.857 | 0.754 | 0.253 | 0.020 |
| | GM-LS-seq | 0.767 | 0.767 | **1.000** | **0.700** | 0.021 |
| | GM-LS-par | 0.8800 | 0.8800 | **1.000** | **0.700** | 0.024 |
| | GREEDA | 0.820 | 0.820 | **1.000** | **0.700** | 0.021 |
| | **MM (Ours)** | **1.000** | **1.000** | **1.000** | 0.308 | **0.010** |

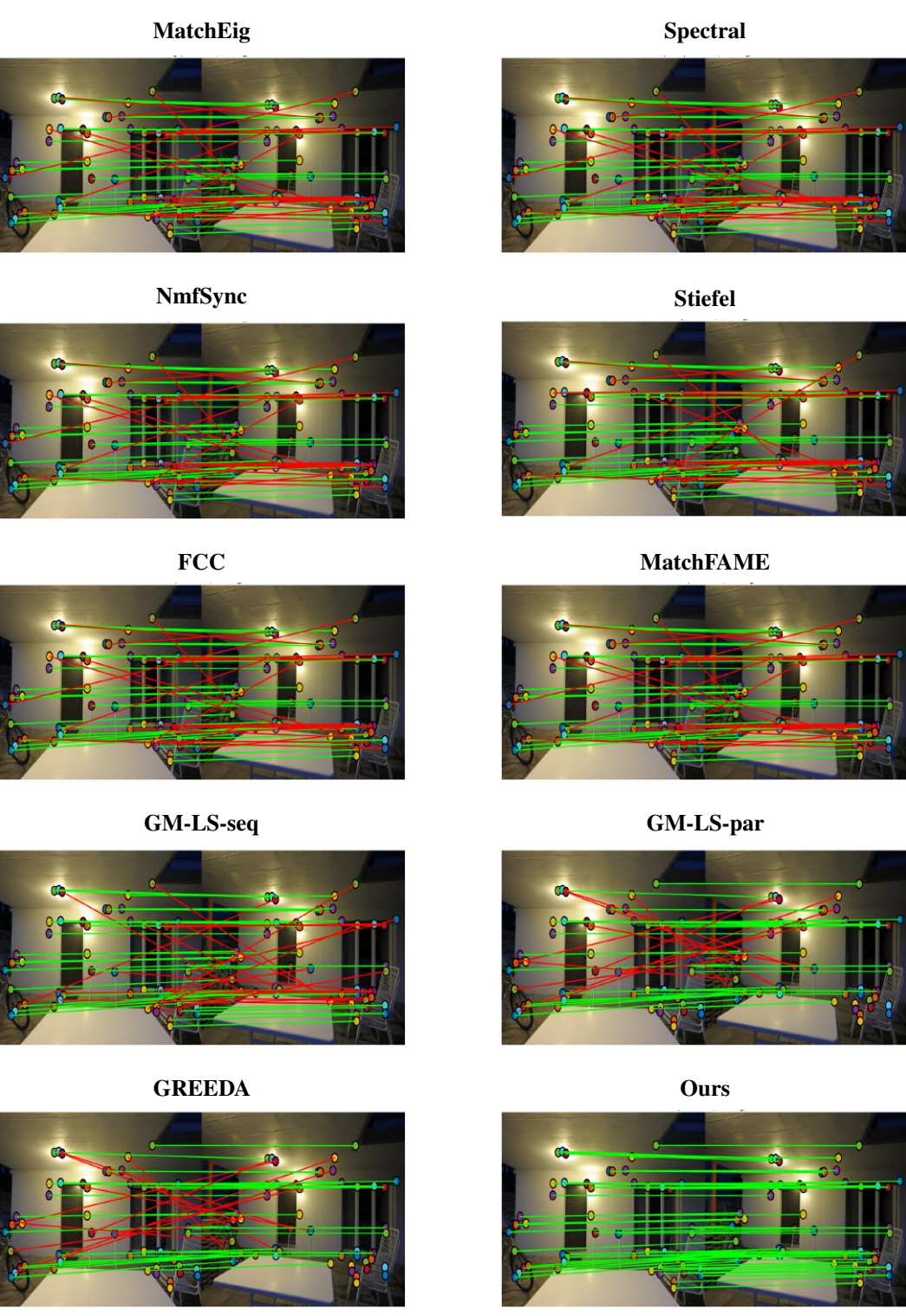

Figure 2: Comparison of matchings between the first and last image of the terrace sequence obtained by different methods.

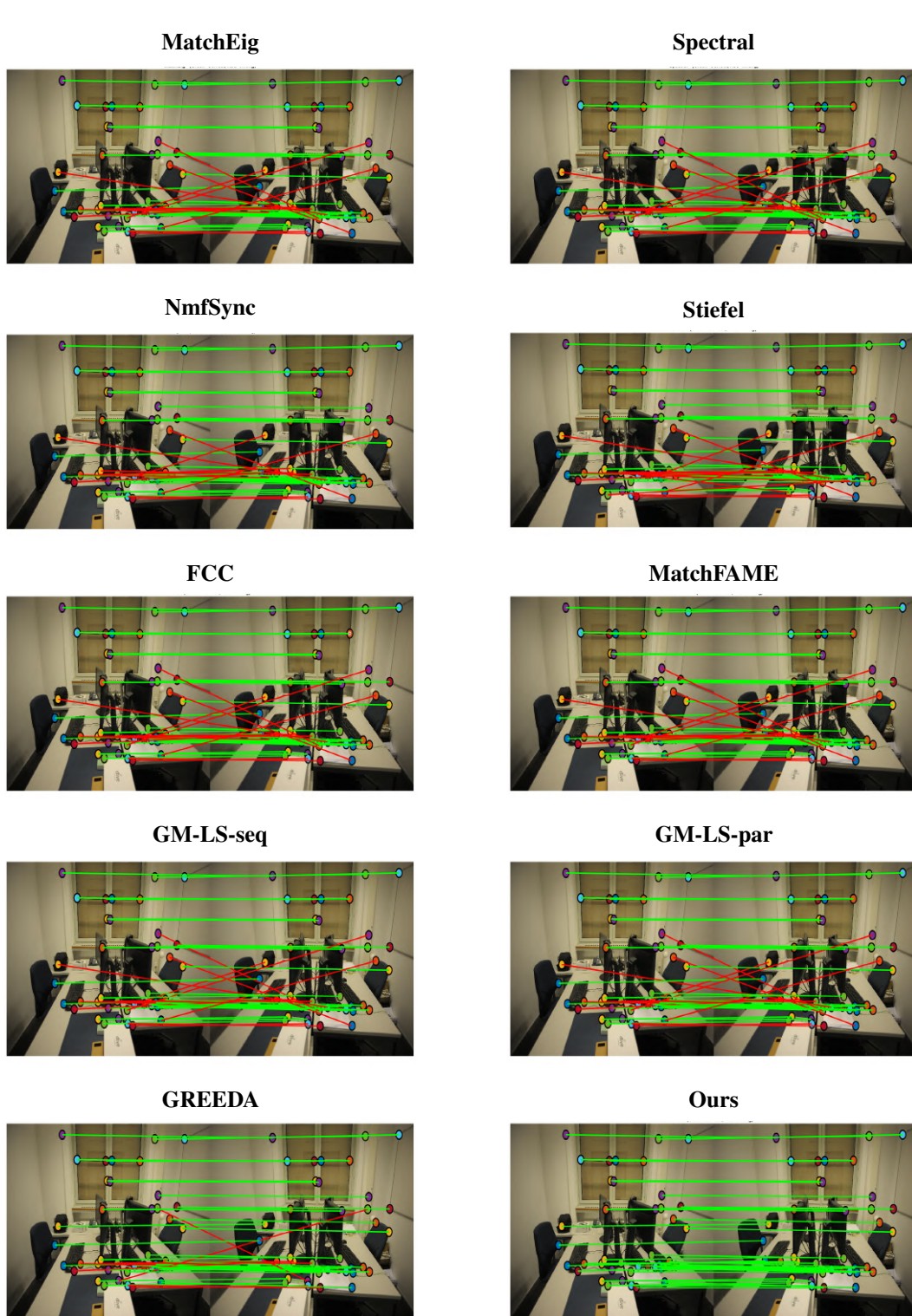

Figure 3: Comparison of matchings between the first and last image of the office sequence obtained by different methods.

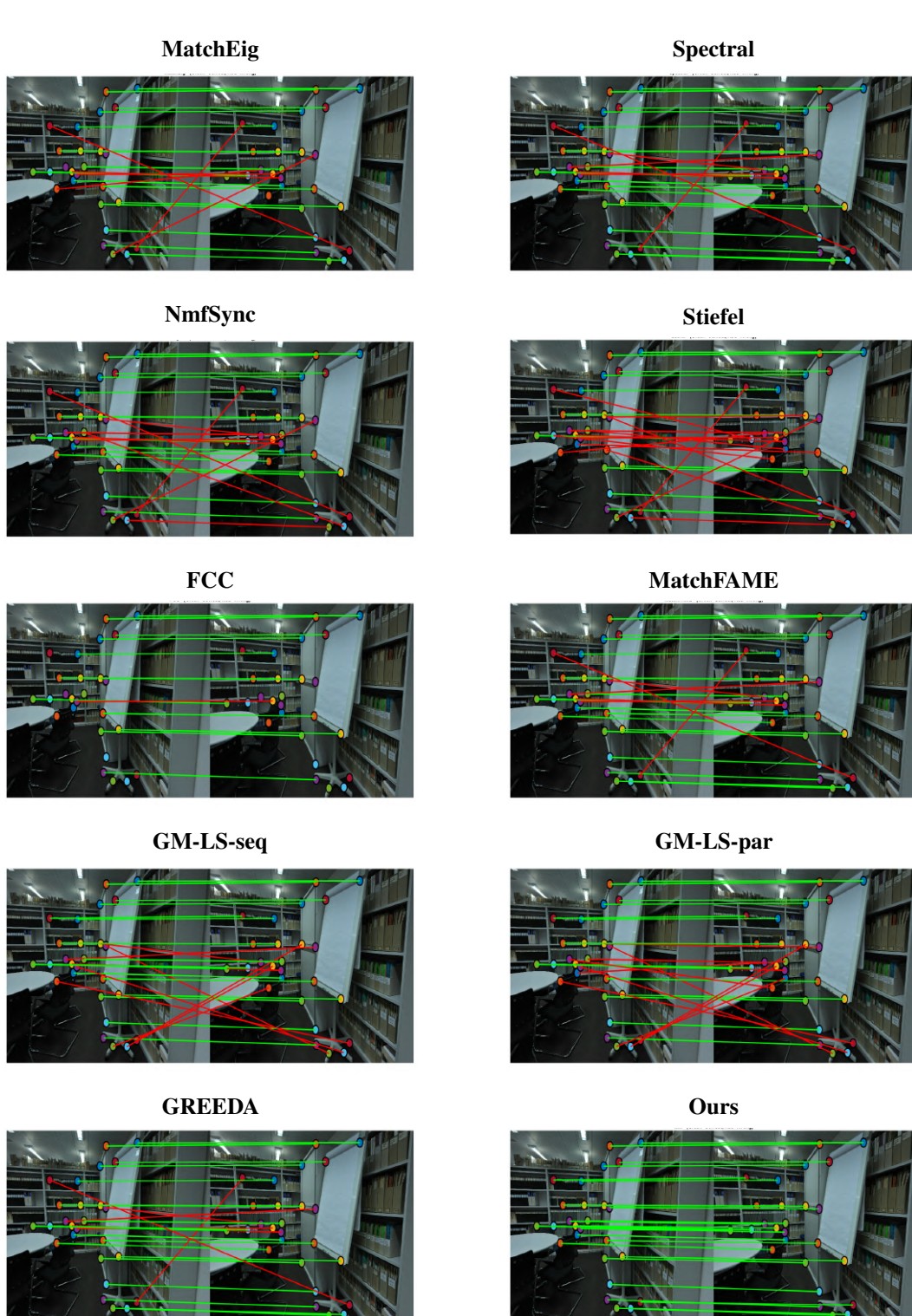

Figure 4: Comparison of matchings between the first and last image of the kicker sequence obtained by different methods.

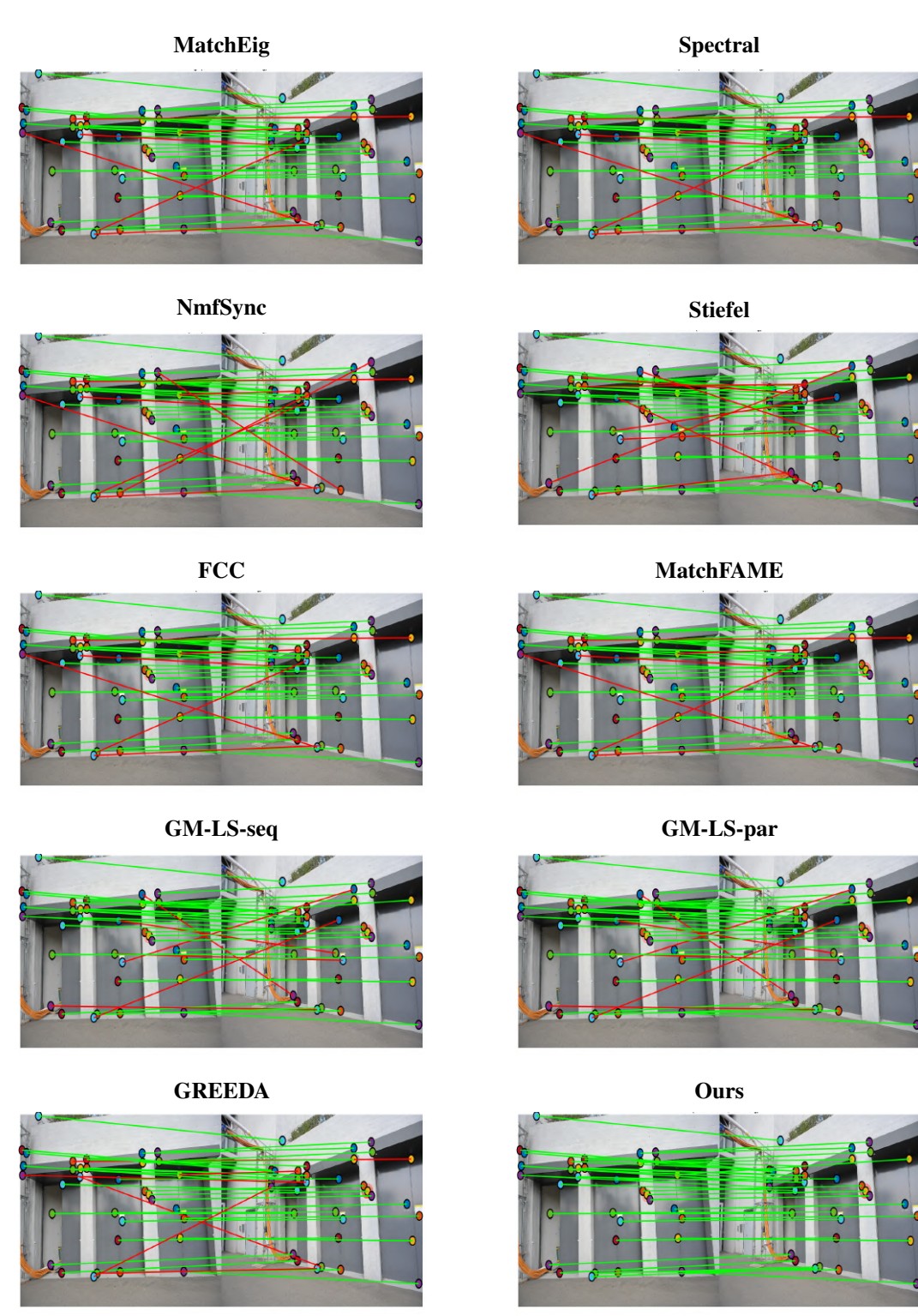

Figure 5: Comparison of matchings between the first and last image of the electro sequence obtained by different methods.

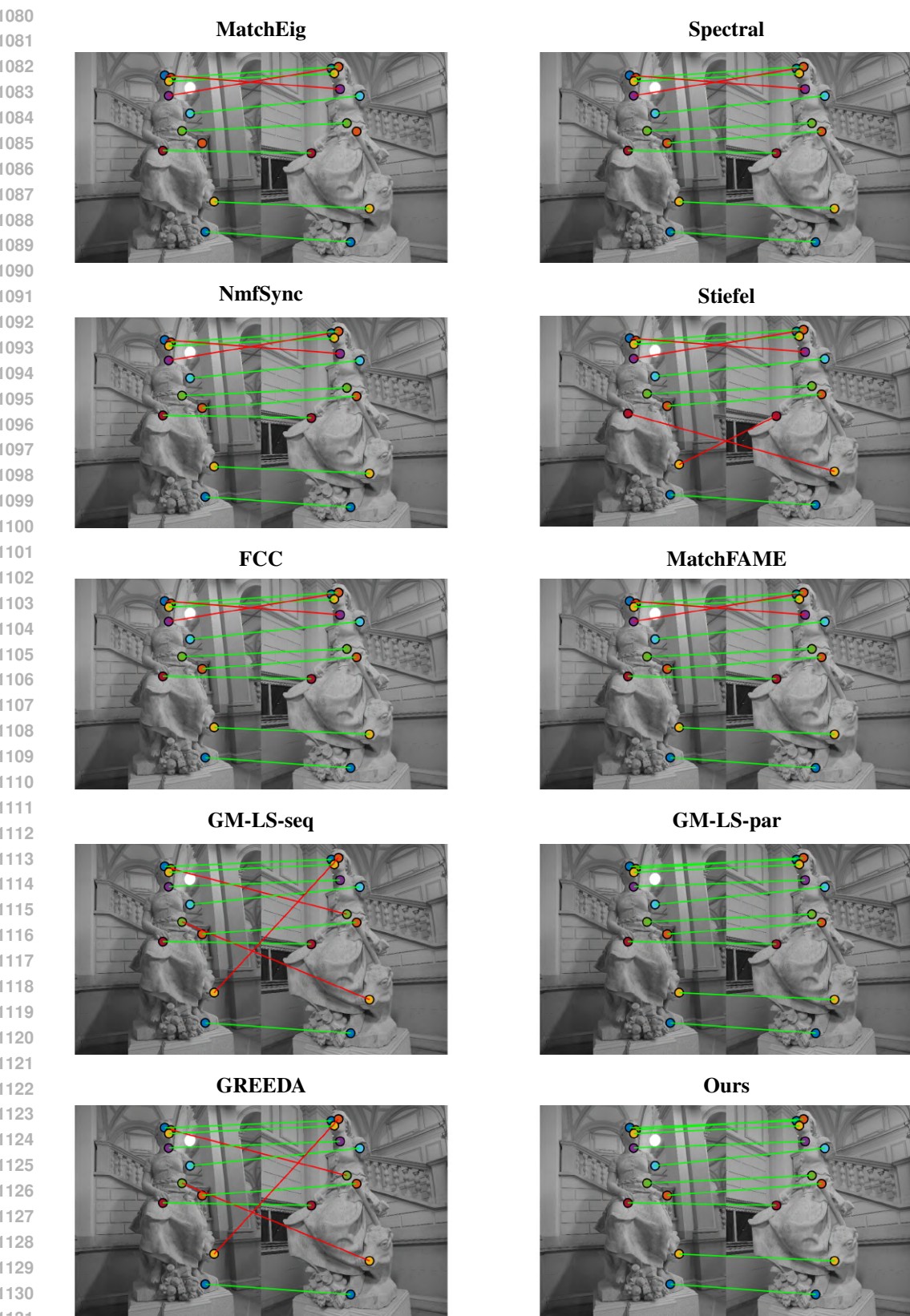

Figure 6: Comparison of matchings between the first and last image of the statue sequence obtained by different methods.

# C COMPARISON OF INITIALIZATION RESULTS

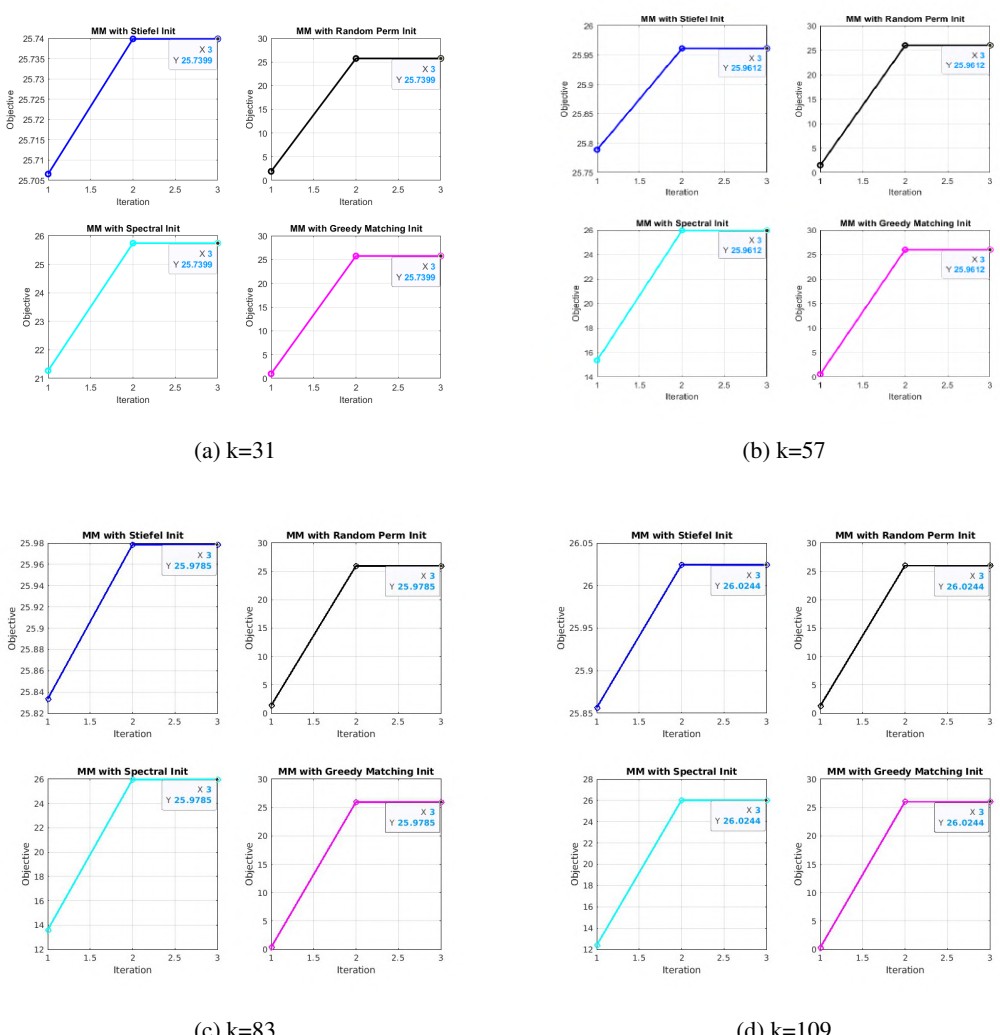

Figure 7: Visualization of results for different initial values.

Due to the non-convex nature of our problem, a theoretical guarantee of convergence to a high-quality maximum independent of initialization is not feasible—a common trait of such optimization landscapes. We therefore provide extensive numerical evidence to demonstrate the practical robustness of our method with respect to the choice of initial parameters.

To this end, we conducted experiments on the CMU House dataset using four distinct initialization strategies: spectral, random permutation, Stiefel, and greedy matching. To ensure that our findings are not an isolated phenomenon, this rigorous comparison was carried out for over forty different odd values of $k$, with the full set of convergence plots publicly available for review[1]. For clarity and consistent scaling across these runs, we adopt the normalization practice of Bernard et al. (2021) and divide each objective function value by $k^2$ in all plots.

Figure 7 illustrates this robust convergence for four representative cases on the CMU House dataset, showcasing the results for $k$ values of 31, 57, 83, and 109. As can be seen, despite their different starting points, all four scenarios consistently converge to the same final objective function value. It is worth noting that this stability extends beyond the objective value; empirical experiments show that the

---

[1]https://github.com/anonymous-user-anonymous-user/
Convergence-vs-iteration

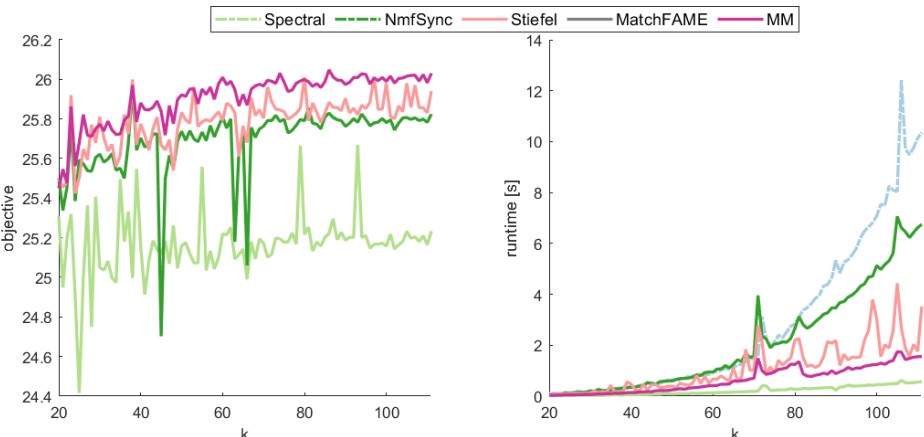

Figure 8: Quantitative results for the CMU house sequence are shown, reporting the objective value of Problem (4) (higher is better). Each point along the horizontal axis represents a different permutation synchronization instance.

resulting matchings for our proposed method are identical for all the initial values we considered. As a result, the performance metrics—F-score, Feature Recall, and cyclic consistency—remain constant across different initializations. We have omitted repetitive tables for brevity, but the implemented code demonstrating identical objective values and maximizer results is provided in the supplementary material. Given the consistent attainment of peak metric values and identical results across the diverse conditions we have considered (e.g., different datasets and initial values), our method empirically demonstrates behavior suggestive of global optimality. However, as the problem is non-convex and NP-hard, a theoretical proof of global optimality is generally intractable Boyd & Vandenberghe (2004). Therefore, while we have proven convergence to a KKT stationary point, we do not claim theoretical global optimality.

As a complementary analysis, we examine the difference between the Stiefel method and our MM-based framework, using the Stiefel solution as a warm-start initialization. As illustrated in Figure 7, when $k = 31$, the objective value difference is $31^2 \times (25.74 - 25.705) = 33.635$. Comparable improvements are observed for $k = 57$, $k = 83$, and $k = 109$, where our framework achieves increases in the objective value of 556.22, 988.57, and 1953.23, respectively. This analysis demonstrates the significant enhancement achieved by the MM-based framework over the Stiefel method when the latter is used as a warm start.

## D  OBJECTIVE VALUE EVALUATION

Figure 8 juxtaposes trace objective and runtime curves for six methods as $k$ increases from 20 to 111.

The objective curve (left): higher values indicate tighter alignment with the ground-truth cycle-consistent structure. It is worth mentioning that the objective plot has been drawn just for methods that deal with the same objective function. MM's objective (magenta) climbs smoothly from around 25.4 at $k = 20$ to about 26.0 at $k = 111$, with negligible jitter, confirming the guaranteed monotonic ascent of our MM iterations. Stiefel (light red) follows closely but exhibits small dips when its local relaxations fail to fully capture newly added permutations, before recovering in subsequent iterations. NmfSync (dark green) achieves a competitive starting objective but shows more pronounced dips and plateaus, indicating occasional surrogate gaps. Spectral (light green) remains the lowest, with significant noise at small $k$ due to eigenvector instability, and only gradually increases thereafter.

In the runtime panel (right), as we can see, spectral (light green) is the fastest, owing to its single-shot eigen-decomposition, but at the cost of lower accuracy. MM (magenta) executes in roughly $1.3\,\mathrm{s}$ at $k = 111$—only marginally slower—by leveraging efficient pivot-based assignment steps and sparse matrix multiplications. Stiefel (light red) takes about $4.0\,\mathrm{s}$, as its iterative relaxations invoke

repeated EVD- or SVD-like steps under the hood. NmfSync (dark green) consumes $8.3$ s, reflecting its factorization overhead.

Together, these two curves in Figure 8 underscore MM's superior balance: it achieves the highest objective values with comparable runtime to the fastest relaxation methods, indicating that MM's exact combinatorial surrogates capture more of the affinity than relaxation-based baselines.

The results in Figure 8 underscore MM's superior performance: it achieves the highest objective values, indicating that MM's exact combinatorial surrogates capture more of the affinity than relaxation-based baselines.

# E    Broader Applications of Our Solver

Beyond permutation synchronization, the proposed exact combinatorial MM optimization framework provides a unified foundation for various discrete inference problems. In particular, three applications are described below, to which our MM-based approach — with minor structural modifications — can be effectively applied.

## E.1    Multi-Model Fitting and Geometric Consensus

A significant challenge in computer vision is multi-model fitting—the task of grouping noisy data points into multiple geometric structures (such as lines, circles, or homographies) while simultaneously rejecting outliers. Recent state-of-the-art approaches, including those exploring Quantum Annealing Pandey et al. (2025), formulate this as a consensus maximization problem. These methods typically map the task to complex binary optimization models to determine which points belong to which model. Our proposed framework can be effectively adapted to this domain. By treating the assignment of points to geometric models as a discrete optimization task similar to matching, our approach offers a highly efficient classical alternative. It is capable of solving the consensus problem deterministically, avoiding the hardware limitations and embedding constraints often associated with quantum-based solvers.

## E.2    Semi-Supervised Learning and Label Propagation

In the field of machine learning, semi-supervised label propagation aims to infer the class labels of a large set of unlabeled data based on a small number of labeled examples and the underlying graph structure. Standard techniques typically relax the discrete class indicators into continuous real values to minimize an energy function involving the graph Laplacian Holtz et al. (2024). This relaxation often leads to ambiguity when mapping the continuous results back to discrete classes. Our framework is naturally suited to address this by maintaining the problem in its discrete form. It can be extended to optimize the label assignment directly on the graph, ensuring that the inferred labels remain valid integers throughout the process, thereby avoiding the errors introduced by continuous relaxation and post-hoc rounding.

## E.3    Quadratic Unconstrained Binary Optimization (QUBO)

A fundamental formulation for many NP-hard combinatorial problems—ranging from Max-Cut and Maximum Independent Set to social network clustering—is the Quadratic Unconstrained Binary Optimization (QUBO) problem. A growing body of research attempts to solve QUBO by transforming the optimization task into a learning problem. These methods typically employ Graph Neural Networks (GNNs) to treat the binary variable assignment as a vertex classification task, using strategies like Batch Greedy Flipping to iteratively refine solutions based on learned probabilities Chen et al. (2024). However, these learning-based methods require extensive training data and often struggle to generalize to graph structures unseen during the training phase. Our proposed exact combinatorial MM framework offers a robust, training-free alternative for this domain. By constructing tight linear surrogates for the quadratic affinity terms, our method reduces the complex energy landscape into a sequence of exact linear steps. This provides a deterministic path to high-quality solutions for general binary optimization problems without the overhead of training neural networks or the unpredictability of stochastic search heuristics.

