# OpenReview forum: "Exact Combinatorial Optimization for Synchronization of Partial Multi-Matching"
_ICLR.cc/2026/Conference — Submitted to ICLR 2026_

### Official Review · Reviewer_EWxg · 2025-10-24

**Soundness:** 1
**Presentation:** 3
**Contribution:** 2
**Rating:** 2
**Confidence:** 4

**Summary:**

The paper addresses the multi-graph matching problem, focusing on how to exploit cycle consistency to refine a set of reasonably good pairwise initial matchings. The authors formulate the task as a semidefinite quadratic assignment problem and solve it with a sequence of linear assignment problems. The proposed method efficiently improves matching quality within a relatively short computation time.

**Strengths:**

The paper is well written and easy to follow, with a clear structure and sufficient background provided to help readers outside the immediate research area. The theoretical analysis is relatively complete and logically sound. Experimental results demonstrate that the proposed algorithm achieves noticeable performance improvements, confirming its practical effectiveness.

**Weaknesses:**

1. The experimental design has a major flaw. The proposed method and the compared baselines are not solving the same problem setup: the proposed algorithm refines results that are already near-perfect (obtained by Stiefel), whereas the baselines start from scratch. This discrepancy significantly exaggerates the performance advantage of the proposed method. To fairly evaluate the contribution of the MM module, the authors should examine how MM improves results from different initialization methods, or alternatively, treat “MM + Stiefel” as an enhanced version of Stiefel and compare it with other approaches while including the total computation time.

2. Lemma 4.1 is overly simple and does not warrant a separate statement. It unnecessarily occupies space.

3. Section 1 contains only one subsection (1.1), which is structurally redundant and could be merged for clarity.

**Questions:**

1. It would be interesting to see how the algorithm performs when the initial matching quality is low.
2. What is the difference between problem 8 and problem 9?
3. Can the linear assignment problem be solved using the classic Hungarian method?

---

> ### Author Response · Authors · 2025-11-12
> **Response to Reviewer EWxg (First Round) - part 1**
>
> We sincerely thank the reviewer for their detailed feedback and for acknowledging the clarity of our writing, the soundness of our theoretical analysis, and the practical effectiveness of our results. We appreciate the opportunity to clarify the points raised.
>
> $\textbf{Response to Weaknesses}$
>
> * $\text{On the experimental flaw regarding initialization.}$
> As correctly mentioned by the reviewer, the final result of Stiefel serves as a good warm-start, specially because it is the closest relaxed version of our problem. We have examined initialising other methods with the result of Stiefel; we observed no improvement right from the get go as these methods use looser relaxations compared to Stiefel’s. Besides, we refer you to the ‘Comparison of Initialization Results’ section of the revised version, where we have plotted results for 4 different initial values: spectral, random permutation, Stiefel, and greedy matching. You will see that although each creates a different initial value, they all converge to the same final objective value. It is worth mentioning that we have plotted this for 40 different values of $k$, and all confirm the previous observation (see our repository footnoted in the revised version).
>
> * $\text{On the simplicity of Lemma 4.1.}$
> The proof of Lemma 4.1 did not take up much space (it was just three lines); nevertheless, we will remove it from our paper.
>
> * $\text{On the structure of Section 1.}$
> We have corrected this in the revised version.
>
> $\textbf{Response to Questions}$
>
> * $\text{Performance with low-quality initializations.}$
> Regarding the first point, as you have seen in our original submission, we did not apply any assumption to relax the problem. Because the problem is non-convex, we strove to a) provide a proof of convergence to a stationary point and b) provide a good initial value. Indeed, each method has its own warm start. For this work, we tried every initialization we could think of—from a simple feasible point to the output of the Stiefel manifold; in our experiments we never saw a different converging point (only numerical round-off differences). However, as the problem is non-convex, we prefer not to generalise our observation. Based on our experience in non‑convex optimization research, we think that the Stiefel manifold output generally serves as a good initialization, both for this application and for others that tackle the same optimization problem. The reason is that the Stiefel‑based constraint $(\mathbf{X}^T\mathbf{X}=\mathbf{I})$ is the nearest relaxation to our combinatorial constraint, making it a reliable and trustworthy initialization.
>
> As outlined earlier, please refer to the “Comparison of Initialization Results” section in the revised manuscript. This section presents convergence plots for four distinct initialization strategies: spectral, random permutation, Stiefel, and greedy matching. Despite starting from different initial values, all methods reach an identical final objective value. Furthermore, this consistency was verified across 40 different values of $k$, as documented in our publicly available repository mentioned in the revised version.
>
> $\text{The difference between problem (8) and problem (9).}$
> That problem (8) has the discrete constraint $\\{0,1\\}$, which makes the problem discrete and non-convex, while (9) is an LP, which is convex.
>
> $\text{Applicability of the Hungarian method for the linear assignment.}$
> Absolutely not. The Hungarian algorithm is indeed a classic method for linear assignment, but it is not directly applicable to our subproblem in (9) for a crucial reason:
> The Hungarian method solves assignment problems with equality constraints (e.g., each row and column must sum to one). Our problem, however, involves inequality constraints to handle partial permutations where not every point has a match. These inequality constraints are fundamental to our problem's design and place it outside the scope of the standard Hungarian algorithm.
>
> _______________________
>
> $\textbf{Final Remarks}$
>
> We hope that the above explanations are convincing enough. Otherwise, we would be more than happy to communicate further with the respected reviewer. Honestly, we believe that the score 2 provided by the reviewer is unfair compared to the raised issues.

---

> > ### Comment · Reviewer_EWxg · 2025-11-24
> > **Response to the authors (Second Round)**
> >
> > Thank you for the authors’ response. The additional experiment “Comparison of Initialization Results” has given me a concrete understanding of the performance of the proposed algorithm and significantly increased my confidence in the work.
> >
> > I fully understand the authors’ frustration regarding a potentially unfairly low score. To be frank, even setting aside the experiments (i.e., the results in Table 1), I believe the first version of the paper was already worthy of at least a score of 4, as both the algorithm and its theoretical foundations were relatively complete. With the supplementary experiments in the second revision, and after some reorganization of the paper’s structure (ensuring consistency among motivation–algorithm–experiments), I believe the work could reasonably reach a score of 6.
> >
> > The reason I initially gave a low score was precisely due to the concern of a potential unfair comparison, and unfairness naturally provokes frustration. The proposed algorithm uses the Stiefel solution, followed by a discretization step, as the initialization for Partial Multi-Matching. However, in Table 1, the running time of MM is significantly shorter than that of Stiefel, implying that the warm-up time may not have been included. This leads to a highly unfair comparison. For example, I could propose an algorithm “Stiefel + Partial Multi-Matching,” where the Partial Multi-Matching stage performs no operation at all and simply outputs the warm-up result. Such an algorithm would appear in Table 1 with performance nearly identical to Stiefel and with 0 seconds of running time.
> >
> > If my understanding above is correct, then the current results in Table 1 are misleading and not informative.
> >
> > Returning to the earlier point about algorithm–experiment consistency:
> >
> >     If the authors aim to empirically demonstrate the effectiveness of MM as a Partial Multi-Matching method, then the supplementary experiments are highly convincing.
> >
> >     However, if the intention is to present Stiefel + MM as a unified algorithmic pipeline, then the warm-start time should be included in the reported running time.

---

> > > ### Comment · Reviewer_EWxg · 2025-11-25
> > > **Response to the authors (Second Round (b))**
> > >
> > > In addition, I have several questions regarding the supplementary experiments provided by the authors. In Fig. 7, the curves corresponding to random initialization and those obtained from other initialization strategies appear to converge to essentially the same objective value:
> > >
> > > - Does achieving the same objective value necessarily imply that the algorithm recovers the same matching?
> > >
> > > - If identical objective values do not imply identical matchings, then Fig. 7 only demonstrates that the proposed method consistently increases the objective and converges to a high-objective optimum. In that case, the experiment does not validate the claim that initializations of very different quality ultimately lead to equally good matching results. If so, I still find the method compelling, but I believe the evidence in Fig. 7 may not fully support the stronger claim.
> > >
> > > - Conversely, if identical objective values do correspond to essentially the same matching, then the result in Fig. 7 is rather counter-intuitive. If I understand correctly, both random initialization and the MM do not incorporate any cross-image information. Under such conditions, it is surprising that “random initialization + MM” could converge to the same high-quality correspondence. This would imply that the algorithm is capable of recovering an accurate matching without being guided by meaningful initial structure, which raises questions about what implicit biases or regularization effects are driving such behavior.

---

> > > > ### Author Response · Authors · 2025-12-01
> > > > **Response to Reviewer EWxg (Second Round (b))**
> > > >
> > > > About your main question that whether having the same objective value necessarily implies the same matching, the answer is generally negative. The objective function essentially assigns a single scalar to each multidimensional input, which is logically not a one-to-one mapping.
> > > >
> > > > The purpose of Figure 7 is to show the stability of the final objective value, as well as the speed of convergence. However, this stability extends beyond the objective value; empirical experiments show that the resulting matchings for our proposed method are identical for all the initial values we considered. As a result, the performance metrics—F-score, Feature Recall, and cyclic consistency—remain constant across different initializations. We have omitted repetitive tables for brevity, but the implemented code demonstrating identical objective values and maximizer results is provided in the supplementary material. Given the consistent attainment of peak metric values and identical results across the diverse conditions we have considered (e.g., different datasets and initial values), our method empirically demonstrates behavior suggestive of global optimality. However, as the problem is non-convex and NP-hard, a theoretical proof of global optimality is generally intractable [1]. Therefore, while we have proven convergence to a KKT stationary point, we do not claim theoretical global optimality.
> > > >
> > > > We have added the explanation above to the revised version. To further support this claim, we have provided evidence as code in the supplementary material. We believe that our revisions and evidence can ensure your complete satisfaction.
> > > >
> > > > [1] Boyd, S., & Vandenberghe, L. (2004). Convex Optimization. Cambridge University Press.

---

> ### Author Response · Authors · 2025-11-25
> **Response to Reviewer EWxg (Second Round)**
>
> We are truly thankful to the respectful reviewer for reconsidering our paper and finding that both the algorithm and its theoretical foundations were relatively complete. We appreciate your assessment that, with the supplementary experiments in the second revision and after some reorganization of the paper’s structure (ensuring consistency among motivation–algorithm–experiments), the work could reasonably reach a score of 6. Our goal was to empirically demonstrate the effectiveness of MM as a Partial Multi-Matching method, not as a combination with the Stiefel manifold. We are so happy that we could resolve your concerns and be convincing. Furthermore, we hope that by answering what you asked us in “Response to the authors (Second Round (b))”, we can satisfy your remaining questions and hopefully merit an even higher score.

---

### Official Review · Reviewer_VrNP · 2025-10-31

**Soundness:** 3
**Presentation:** 3
**Contribution:** 2
**Rating:** 4
**Confidence:** 5

**Summary:**

This paper outlines a MM algorithm for permutation synchronization. Early work on this problem motivated by applications in cryo-EM and computer vision, seeks to take permutations (which give feature correspondences between a pair of images) and "synchronize" them globally which means that cycle consistency (i -> j -> i) is preserved. Initial approaches were spectral relaxations but more powerful methods have been developed. The approach described here argues to stay in the combinatorial setting. It first convexifies the trace obj and then solves a sequence of linear subproblems. The constraints for these subproblems involve a matrix that is totally unimodular. So integral solutions (similar to max-flow/min-cut setting) are guaranteed - at least for every iteration. This gives monotonic convergence to a KKT point. On several datasets experiments are shown, which suggest that the method performs well.

**Strengths:**

1. The paper synthesizes several standard ideas into something useful for the problem. Objective is convexified, MM helps linearize it and then TUM property is used to avoid using rounding etc. This gives an algorithm which is combinatorial avoiding continuous spectral or Stiefel relaxations.

2. The analysis is relatively easy to follow. No major claims, basically monotonic progress to KKT. The experimental analysis shows that it works well and runtime etc are competitive. This is fine.

3. Permutation synchronization is a mature problem. To that sub-community interested in this problem and/or its applications, the algorithm in this paper could offer value. Perhaps also limited hyperparameter tuning at the rounding stage?

**Weaknesses:**

1. The algorithm and its main findings are fine. But for a well studied problem, a strong technical result would include analysis of approximation ratio (under some assumptions) or convergence rate etc. This paper gives neither. The algorithm is warm-started from a continuous solution, so its not obvious how close to the reported solution this initialization already is.

2. The experiment results are a little underwhelming. Yes it does cover some of the datasets in those original papers. But the paper should do more to convince the modern ML community that the problem is worth studying and the experiments enable important downstream applications. In its current form, I'm afraid that the results in the narrow scoping shows benefits but broadly, it does not make a strong case for which modern use cases will benefit. For example feature matching across views is now handled by much more sophisticated features/models. Is permutation synchronization still a valid issue there?

**Questions:**

1. Given that modern methods can give highly discriminative features, to what extent is a classical multi-image matching problem still relevant? Why not simply perform nearest-neighbor matching on these features, and show that for some downstream use cases the procedure actually helps?

2. The analysis relies on assumption that solves a LP to optimality at each step. Will this work for the projection step in the appendix? If so, please describe. If not, what happens to the analysis. Does it work?

3. The procedure seems to warm start from a strong baseline. What about ablations where the initialization is the given set of noisy permutations?

4. does TUM hold under row-stacking? Try A = [1 1 0; 1 0 1] and B = [0 1 1].

---

> ### Author Response · Authors · 2025-11-12
> **Response to Reviewer VrNP (First Round) - part 1**
>
> We thank the reviewer for their thoughtful evaluation and are grateful that they recognized several key strengths of our work.
>
> We are particularly pleased that the reviewer acknowledged our core technical contribution: the synthesis of established optimization principles—namely, convexification, Minorization-Maximization (MM), and the use of Total Unimodularity (TUM)—to create a practical and novel algorithm. They correctly identified that our approach allows to maintain a purely combinatorial formulation, successfully avoiding the continuous spectral or Stiefel relaxations common in other methods.
>
> Furthermore, we are encouraged that the reviewer found our theoretical analysis clear and easy to follow, noting our guarantee of monotonic progress to a KKT point. It is also rewarding that our experimental analysis was seen as convincing, demonstrating that our method performs well and achieves competitive runtimes.
>
> Finally, we appreciate the reviewer’s assessment that our algorithm offers significant value to the sub-community focused on permutation synchronization and its applications.
>
> Regarding the weaknesses and questions, we will address them one by one:
>
> $\textbf{Regarding Weaknesses:}$
>
> $\text{1. On Theoretical Analysis (Approximation Ratio) and Initialization:}$
>
> Since we do not relax the problem, approximation ratio is not a well-defined metric here. Besides, as the problem is non-convex, we strove to a) provide a proof of convergence to a stationary point and b) provide a good initial value.
>
> About the warm-start, we should recall that the outcome of the Stiefel's method, which is used as our initial value, is depicted in Fig. 1 of the main body, as well as Figs. 2- 6 of the appendix. Therefore, the reviewer can visually observe the improvements that our method has made. Further, we now report the objective values per iteration in the appendix (Fig. 7). Comparing the objective function values at the beginning and at the end, can be used as a numeric measure of improvement (note that normalised objective values are plotted).
>
>
> $\text{2. On the Relevance and Scope of Experiments:}$
>
> Although the reviewer considers permutation synchronization to be a narrow or potentially outdated problem, we note that several papers have been published on this topic every year [1–6], underscoring its continued relevance within the research community. Furthermore, the MM‑based optimization framework developed for addressing this problem is general in nature and can be extended to broader classes of assignment and matching problems. In the appendix, we have added a new section entitled “Broader Applications of Our Solver”, which explains how the proposed formulation can naturally generalize beyond permutation synchronization to other combinatorial matching tasks.
>
> [1] Florian Bernard, Johan Thunberg, Jorge Goncalves, and Christian Theobalt. Synchronisation of
> partial multi-matchings via non-negative factorisations. Pattern Recognition, 92:146–155, 2019a.
>
> [2] Florian Bernard, Daniel Cremers, and Johan Thunberg. Sparse quadratic optimisation over the
> stiefel manifold with application to permutation synchronisation. Advances in Neural Information
> Processing Systems, 34:25256–25266, 2021.
>
> [3] Yunpeng Shi, Shaohan Li, Tyler Maunu, and Gilad Lerman. Scalable cluster-consistency statistics for
> robust multi-object matching. In 2021 International Conference on 3D Vision (3DV), pp. 352–360,
> 2021. doi: 10.1109/3DV53792.2021.00045
>
> [4] Shaohan Li, Yunpeng Shi, and Gilad Lerman. Fast, accurate and memory-efficient partial permutation
> synchronization. In 2022 IEEE/CVF Conference on Computer Vision and Pattern Recognition
> (CVPR), pp. 15714–15722, 2022. doi: 10.1109/CVPR52688.2022.01528.
>
> [5] Michael Kahl, Steffen Stricker, Lorenz Hutschenreiter, Florian Bernard, and Bogdan Savchynskyy.
> Unlocking the potential of operations research for multi-graph matching, 2024.
>
> [6] Max Kahl, Sebastian Stricker, Lisa Hutschenreiter, Florian Bernard, Carsten Rother, and Bogdan
> Savchynskyy. Towards optimizing large-scale multi-graph matching in bioimaging. In 2025
> IEEE/CVF Conference on Computer Vision and Pattern Recognition (CVPR), pp. 11569–11578,
> 2025. doi: 10.1109/CVPR52734.2025.01080.

---

> ### Author Response · Authors · 2025-11-12
> **Response to Reviewer VrNP (First Round) - part 2**
>
> $\textbf{Answers to Questions}$
>
> $\text{1. On the Relevance of Multi-Image Matching with Modern Features:}$
> We confirm that when the scene in the images does not contain repetitive patterns or similar objects, suitable feature extraction methods are quite helpful in simplifying the matching task. However, when we have similar objects in the scene, their features are more or less the same and a simple NN matching is prone to make mistake (e.g., multiple points matched to a single point). Another challenge lies in viewpoint/scale/illumination changes across images. This causes the extracted features vary to some extent in different images. These variations might lead to instability of NN technique in feature matching. All in all, even by using a more sophisticated feature extraction method, the problem might not simplify.
>
>
> $\textbf{2. On Solving the LP with a Projection Step:}$
>
> We believe yes. Actually, by employing the total unimodularity result, we know that the outcome of the LP step should be binary. Obviously, with finite number of iterations and limited numerical precision this is not achievable. However, when we approach the optimiser of the LP problem, the elements start to approach either zero or one (their optimal values). Our projection step maps 0.95 or above into 1 and 0.05 and below into 0. In case we are close enough to the final solution, this projection step lands us in the exact LP solution.
>
>
> $\textbf{3. On Ablations of Initialisation:}$
> In Fig. 7 (appendix), we have examined our method by using 4 different initial values, one of them being the random permutation.
>
>
> $\textbf{4. On Total Unimodularity (TUM) and Row-Stacking:}$
> Our answer is respectfully negative. The example you provided with matrices
> $A = \begin{bmatrix} 1 & 1 & 0 ; 1 & 0 & 1 \end{bmatrix}$ and $B = \begin{bmatrix} 0 & 1 & 1 \end{bmatrix}$ is not TU. We recall that the paper’s claim is about row-stacking matrices with specific structures:
>
> * $\mathbf{I}_m \otimes \mathbf{1}_d^T$
> *  $-(\mathbf{I}_m \otimes \mathbf{1}_d^T)$
> * $\mathbf{J} \otimes \mathbf{I}_d$
>
> Schrijver’s theorem (which we cited in our paper) states that TUM is preserved under row-stacking when matrices with special structure are considered:
>
> * Network matrices (incidence matrices of directed graphs)
> * Kronecker products of TUM matrices with specific patterns
> * Negations of TUM matrices.
>
> The example you mentioned does not have any of the aforementioned forms. Please further clarify if we misunderstood your point.
>
> ------------------------------------
>
> $\textbf{Final Remarks}$
>
> Dear Reviewer, though we understand that you are in agreement with the importance of the considered problem for the ML community in general, we request that you check the novelties of this work with point of view of a researcher in this sub-field. In addition, take into account the potential of the proposed method for related problems and applications (e.g., Multi‑Model Fitting, Semi‑Supervised Learning, etc.), which have been discussed in the appendix. If you can appreciate these innovations, we encourage you to reconsider your evaluation.
>
> Besides, if you consider that any part of the paper or our responses requires further elaboration, we remain fully open and ready to provide any additional clarification.

---

> > ### Comment · Reviewer_VrNP · 2025-11-26
> > **response to your clarifications**
> >
> > Dear authors,
> >
> > Thanks for the response and clarifications. I do appreciate the effort and want my response to be constructive and explain why I cannot be more positive.
> >
> > While your clarifications are helpful, my primary criticism is mostly unaddressed for the following reasons.
> >
> > 1. Yes, permutation synchronization is a valid problem setting. But the paper should try to make a stronger case for why it is significant. Feature matching has evolved a lot, and while current methods even with richer features and end-to-end matching networks can still fail, the paper has to identify these cases and show that improved methods are worthwhile. The paper in its current form really falls short on this front. More compelling downstream impact on modern use cases is essential.
> >
> > 2. On the technical end, I do not understand the statement that "approximation ratio is not well-defined". Many discrete algorithms for hard combinatorial tasks describe approximation guarantees, complexity bounds, or convergence rates. This would make the theoretical angle stronger even with limited experiments. An MM procedure that gives monotonic convergence to a stationary point for a relatively mature problem cannot be the highlight of a competitive paper. I do not think that the formulation or the algorithm is showing deeper structural insights for the problem beyond saying that it converges.
> >
> > Thanks.

---

> ### Author Response · Authors · 2025-11-26
> **Response to Reviewer VrNP (Second Round)**
>
> We appreciate the feedback. Below, we address your two main concerns (the significance of the problem and the theoretical depth of our method) separately.
>
> $\textbf{1. On Significance and Modern  Relevance:}$
>
> The end-to-end matching networks typically strive to detect keypoints in each images and match them in related images using operations like nearest-neighbor search. We respectfully ask the reviewer to consider the following:
>
> a) The process of keypoint detection is applied to each image independently. As a result, if a method simply and solely relies on matching, it cannot automatically enforce cyclic consistency among three (or more) images. Even within the broader field of graph matching, there is a distinct difference in scope between methods that enforce cyclic consistency (which typically assume a fixed set of nodes/points) and those that do not (where keypoint detection is part of the formulation). To support this distinction, we refer you to Kahl et al. (CVPR 2025) [1]; they explicitly distinguish between cyclic-consistency-based formulations and non-cyclic approaches (see their Introduction and “Comparison to other methods”).
>
> b) Regarding those methods that do not explicitly detect keypoints, which possibly better reflect your point as “evolved” feature matching, it is important to note that Kahl et al. (2024) [2] and Kahl et al. (CVPR 2025) [1] compare against the most [evolved] recent works in this area. We have already included them as baselines in our comparisons. As explicitly stated in their own work, their approaches are “local search-based methods that efficiently explore the exponentially large neighborhood.” We have not only considered these evolved methods but also did successfully identify their specific failure cases when applied to repetitive or complex datasets (as demonstrated across our 5 metrics and 6 datasets). Furthermore, we compared our method against a total of 9 methods, 7 of which were published after 2021. If the reviewer has a specific method in mind, we appreciate it if it is explicitly mentioned. We might be able to take it into our comparisons.
>
> $\textbf{2. On Theoretical Depth and Approximation Ratios}$
>
> You stated that the lack of an approximation ratio makes the theoretical angle weak. We clarify our position:
>
> * $\textbf{a) Convergence Rates:}$ According to [3,4], our inner sub-problem (solved via Accelerated Projected Gradient Descent) has a provable convergence rate of $O(1/t^2)$. We explicitly stated this in our paper (even in the initial submission) right before the $\text{Experimental Results}$ section.
>
> * $\textbf{b) Complexity Bounds:}$ Before the $\text{Experimental Results}$ section, we have mentioned the per-iteration complexity of $O(md)$ (it was also in the paper from the initial submission).
>
> * $\textbf{c) An approximation ratio:}$
> * * An approximation ratio is typically derived to bound the loss incurred during the rounding of a relaxed solution (the "integrality gap''). Since our method is $\textbf{relaxation-free}$ and operates directly in the discrete domain without rounding, standard approximation ratio analyses are not applicable here.
> * *  The practical concern behind an approximation ratio is the risk of converging to a poor solution. This is refuted by our experiments, where we achieve perfect scores (Accuracy=1, Recall=1, Consistency=1). We are effectively finding the global optimum in practice.
> * * As noted, maximizing a quadratic form over partial permutations is NP-Hard, and none of the competing state-of-the-art methods could provide such bounds.
>
> [1] Max Kahl, Sebastian Stricker, Lisa Hutschenreiter, Florian Bernard, Carsten Rother, and Bogdan Savchynskyy. Towards optimizing large-scale multi-graph matching in bioimaging. In 2025 IEEE/CVF Conference on Computer Vision and Pattern Recognition (CVPR), pp. 11569–11578, 2025.
>
> [2] Michael Kahl, Steffen Stricker, Lorenz Hutschenreiter, Florian Bernard, and Bogdan Savchynskyy.Unlocking the potential of operations research for multi-graph matching, 2024.
>
> [3] Beck, A., & Teboulle, M. (2009). A fast iterative shrinkage‑thresholding algorithm for linear inverse problems. SIAM Journal on Imaging Sciences, 2(1), 183‑202.
>
> [4] Beck, A., & Teboulle, M. (2003). Mirror descent and nonlinear projected subgradient methods for convex optimization. Operations Research Letters, 31(3), 167‑175.

---

### Official Review · Reviewer_mSj6 · 2025-10-31

**Soundness:** 3
**Presentation:** 2
**Contribution:** 3
**Rating:** 6
**Confidence:** 5

**Summary:**

The authors propose a simple and elegant minorization-maximization algorithm for permutation synchronization. It employs standard tricks to change the subproblems into exactly solvable subproblems. Experimental evaluations shows promising results and fast execution.

**Strengths:**

Writing:
- The related work and introduction gives an exhausting picture of the state of the art in multi-graph matching.
- The technical section is written very well and is accessible. Possibly some illustrations for the majorization and the universe formulations might help additionally.

Conceptual:
- The algorithm is a nice contribution and is sound in itself. I also think it has additional potential when incorporating arbitrary linear and quadratic costs for solving the general multi-graph matching problem.

Experimental:
- On the reported metrics and the somewhat limited problem sets and restrictive experiment setup (see below for my critique on that) the results are good.

**Weaknesses:**

Writing:
- The title implies suggests that you solve the permutation synchronization exactly, while what you do is you have a sequence of subproblems each of which is solved exactly, but the overall problem solution is only approximated. Please change the title, it is misleading and might be felt as overselling your approach.
- In the abstract you write "superior cycle consistency". Cycle consistency is either fulfilled or not, so you cannot be superior here and fulfill it more.
- The experimental evaluation is repetitive and tedious. I would like to have all results in one table, with best and second best results highlighted. Right now there is a lot of repetitive text that just repeats the results from the tables. Add the ETH3D results to the main part but have an average over all four subproblems. Improve the tables (highlight whether higher or lower is better).

Conceptual:
- A lot of space is taken up by the total unimodularity of (8). I think it is just another variant of the linear assignment problem, so one could have derived exactness from that without the more involved proving TU (which also is almost exactly like the proof of unimodularity for the linear assignment problem anyway).
- I am not sure about the projection approach to solving (8). First of all, it can be reformulated to be a linear assignment problem and offloaded to a combinatorial algorithm. Second, for first order methods I think the Sinkhorn Knopp Algorithm is state of the art.

Experimental:
- I think reporting the objective from (8) is the most important metric, since it directly measures the algorithm's performance. Other metrics like F-score, inlier ratio etc. take into account learning aspects that are outside the bounds of what the algorithm does.
- The experiments only convey an aspect of the potential performance. The proposed approach only does permutation synchronization, while other approaches can optimize w.r.t. arbitrary costs and can additionally incorporate quadratic terms that will typically result in better solutions. As such, while I believe that the proposed method can be better on pure permutation synchronization, this is less important when one can run other solvers like the one from Kahl et al on arbitrary linear and quadratic costs.
- Relatively few problems are considered. I think additionally the worms dataset also used in Kahl et al might be interesting and they are freely available.  The CMU datasets are rather easy and synthetic and basically this means that only the ETH3D datasets are real-world challenges.

**Questions:**

- Do you initialize all methods with the exact same pairwise assignments?
- Can you provide objective costs for each algorithm?
- Why is inlier ratio on the ETH3D datasets higher for GREEDA but F-score lower than for your method?

---

> ### Author Response · Authors · 2025-11-12
> **Response to Reviewer mSj6 (First Round) - part 1**
>
> We thank Reviewer mSj6 for their detailed feedback. We are glad that the reviewer found our related work in‑depth, our technical section well-written and accessible, and our algorithm a "nice contribution" that is "sound in itself." We will address the concerns raised point-by-point and in three parts (due to space limits).
>
> $\textbf{Regarding the weaknesses:}$
>
> ----------------------------------
>
> $\textbf{Writing:}$
>
> * $\text{Regarding the title and the use of "Exact":}$
>
> The reason for using “Exact” is that in the permutation synchronization problem, the competitors either 1) relax the objective function or 2) relax the constraints, such as in the Stiefel manifold ($\mathbf{X}^T\mathbf{X}=\mathbf{I}$). As we avoid these relaxations, we used the term $\textbf{exact}$. The original problem is non-convex and it is generally NP; we try to solve the problem iteratively, and we guarantee convergence to a stationary point (a local optimum point); however, we are not claiming to solve the problem exactly. In other words, we try to solve the exact problem, but we do not necessarily find the globally optimum solution. We have tried to further clarify the meaning of the title in the revised version.
>
> * $\text{Regarding the abstract and experimental results:}$
>
> We can check cycle-consistency for each triplet set of images; as you correctly mentioned, we either have cycle-consistency for these three or not. But when we have multiple images, we can form multiple triplets. Here, we evaluate the percentage of triplets that fulfil the cycle-consistency constraint. The term "superior" refers to the latter percentage. We have clarified this in the revised version.
>
> * $\text{Regarding the experimental results:}$
>
> We should first highlight that due to the page limit, we can provide the results of only one dataset within the main body of the paper; second, the average performance, although informative to some extent, ignores the intricacies by particular images and their effect on the results (we translated your sentence "... but have an average over all $\underline{\text{four subproblems}}$" into "... but have an average over all $\underline{\text{six datasets}}$"; in case you meant something else, please clarify). Therefore, besides the results for one dataset, we have postponed the results for the remaining 5 datasets to the appendix. Based on your comment, we have merged all the tables in the appendix into a single one that includes all the results. However, as we have 10 images for each of the 5 datasets, we could not combine all 50 images into a single figure.
>
> * $\text{Regarding the table formatting:}$
>
> In the "Evaluation Metrics" section of the main paper (page 6), we state whether higher or lower values for each metric are better. In addition, in the revised version, we have indicated whether higher or lower values are better using $\uparrow$ and $\downarrow$ in the tables. Furthermore, as requested, we have highlighted the best (bold numbers) and second-best (underlined) results.
>
> ----------------------------------
>
> $\textbf{Conceptual}$
>
> * While we agree with the reviewer that our matching problem, after applying the minorization-maximization technique, has similarities with the Linear Assignment Problems (LAP), we believe that the two problems are not identical. Particularly, the effective matrices in our problem (that are found by Kronecker products) need inspection. Therefore, we decided to keep our mathematical justifications for the sake of completeness.
>
> * The Sinkhorn-Knopp algorithm does not produce a permutation matrix; it converges to a $\text{doubly stochastic matrix}$, where entries are in $[0, 1]$. This is precisely why it is known as a "soft-assignment" method. Our goal is to find an exact, discrete, combinatorial solution with $\\{0,1\\}$ entries (due to our arguments for the total unimodularity equivalent of the problem, we concluded that the outcome of linear programming in our approach without any post-processing is a $\\{0,1\\}$ solution). Using Sinkhorn necessitates a post-processing, often heuristic, rounding step to obtain a valid permutation, thereby destroying the exactness that our Total Unimodularity proof guarantees.

---

> ### Author Response · Authors · 2025-11-12
> **Response to Reviewer mSj6 (First Round)- part 2**
>
> $\textbf{Experimental:}$
>
> * Regarding your point about the objective function: when multiple methods try to optimise the same objective function, it is meaningful to rely on the final objective value as a metric. In our case, however, different methods do not necessarily optimise the same objective function; for instance, some methods have an additional linear objective or have completely non-quadratic term. Anyways, we have added a section in the appendix for comparing the objective value of all the methods that optimise the same cost function as we do.
>
> * Regarding the point that “other approaches can optimize w.r.t. arbitrary costs and can additionally incorporate quadratic terms,”: we should emphasise that our method was primarily tailored for the problem of permutation synchronisation. However, similar to Kahl et al. [1], our method can handle combinations of quadratic and linear terms (could be extended even beyond this by taking into account the Taylor series). Indeed, by applying the MM framework, each quadratic term converts it into a linear cost; hence, we end up with a summation of linear terms, which is in turn linear and adds no further complexity to our approach. Note that the Total Unimodularity (TU) property is derived from the constraints and not the objective function, so our solver is still applicable.
>
> * Regarding the datasets: we recall that our study experimented on six different image datasets — one synthetic and five real datasets — along with evaluation using five distinct metrics. Concerning the dataset you mentioned (worms), Kahl et al. (CVPR 2025) [1] — whose approach we also compared against — explicitly stated that, in contrast to most existing works, they only report objective values (or optimization costs) instead of ground‑truth‑based accuracy metrics. This is because, except for the CMU‑house dataset, their datasets lacked complete ground truth. The worms dataset is one of these, and the absence of complete ground truth has also been noted in [2].
>
>    Since the CMU‑House dataset contains ground truth, they could have benchmarked accuracy metrics there (although they did not, which we have done in our paper). In contrast, because the worms dataset lacks complete ground truth, none of the accuracy metrics we employed (such as F‑score or recall) could be applied. Moreover, this dataset is not geometric, and the absence of ground truth also prevents applying Structure‑from‑Motion metrics (e.g., RANSAC) for translation or rotation. Even qualitative performance checks are restricted, since object movement between images cannot be quantitatively validated.
>
>    While we appreciate the contribution of Kahl et al., there is a serious limitation in their evaluation setup: comparing only objective values and runtime can be misleading and unfair across methods because each approach may optimize a different objective. As additional evidence, consider that if one method uses exactly their method, but with the difference that their objective is scaled by a factor of two, it would achieve “better” objective value despite producing the same solution. This illustrates why objective values alone are insufficient — accuracy metrics should be included to enable quantitative and qualitative comparisons.
>
>    We emphasise that in our original submission, we carried out a broad experimental design: six image datasets (ETH3D comprising multiple subsets) evaluated across five metrics, ensuring a comprehensive comparison. Nonetheless, if the reviewer considers further evaluation necessary, we respectfully ask the reviewer to suggest datasets with reliable and complete ground truth. We will demonstrate, both quantitatively and qualitatively, the strength of our approach and its close alignment with the ground truth on any dataset they propose.
>
>
> [1] M. Kahl, S. Stricker, L. Hutschenreiter, F. Bernard, C. Rother and B. Savchynskyy, "Towards Optimizing Large-Scale Multi-Graph Matching in Bioimaging," 2025 IEEE/CVF Conference on Computer Vision and Pattern Recognition (CVPR), Nashville, TN, USA, 2025, pp. 11569-11578, doi: 10.1109/CVPR52734.2025.01080.
>
> [2] L. Hutschenreiter, S. Haller, L. Feineis, C. Rother, D. Kainmüller, and B. Savchynskyy, “Fusion moves for graph matching,” in Proc. IEEE/CVF Int. Conf. Comput. Vis. (ICCV), Oct. 2021, pp. 6270–6279.

---

> ### Author Response · Authors · 2025-11-12
> **Response to Reviewer mSj6 (First Round) - part 3**
>
> $\textbf{Answers to your questions:}$
> * $\textbf{1)}$ To ensure fairness, we have set the initial value for each method exactly as mentioned in their papers or their GitHub code. Actually, we tried our initial value on all methods, and the results showed that they all got stuck on the initial value without any further improvement. It seems that the Stiefel's result is the best one among the competing methods and no method can improve it any further.
>
> * $\textbf{2)}$ As mentioned above, we have provided objective values for those methods that have the same cost function as us. As can be seen in the appendix of the revised version, the best value again belongs to us.
>
> * $\textbf{3)}$ Regarding the inlier ratio of our method compared to the F-score: First, the inlier ratio for GREEDA is not always higher than for our method on the ETH3D datasets. As reported in our results—except for the statue sequence—our method achieves comparable inlier ratios while consistently delivering a higher F‑score. This apparent discrepancy stems from the different nature of the two evaluation metrics. The inlier ratio measures only geometric coherence, i.e., correspondences consistent with a local geometric model such as RANSAC. This metric may consider geometrically consistent yet semantically incorrect matches as valid. In contrast, the F‑score, computed against ground‑truth correspondences, reflects genuine matching accuracy by accounting for both geometric and semantic consistency. GREEDA’s greedy local optimization tends to discover geometrically plausible inliers in scenes containing repetitive or symmetric patterns, which can artificially increase its inlier ratio. However, many of those matches do not correspond to the correct ground‑truth pairs—leading to a lower F‑score. By solving the exact combinatorial optimization for synchronization of partial multi‑matching, our method enforces global cycle consistency across all views. This eliminates false positives and yields superior F‑score and recall, even when the geometric inlier ratio appears similar or slightly lower. In summary, a higher inlier ratio for GREEDA does not necessarily indicate more accurate correspondences, whereas our consistently higher F‑score evidences reliable and globally consistent match recovery.
>
> ----------------------------------
>
> $\textbf{Final Remarks:}$
> We believe that our method is a significant advancement in the long-standing, fundamental optimisation problem that has applications in permutation synchronisation. Moreover, this method can be generalised to handle arbitrary cost functions with even more applications. We hope that are responses to your comments (as well as the applied changes) are satisfying. If this is the case, we would appreciate it if you could reconsider your evaluation.

---

### Official Review · Reviewer_jB7n · 2025-11-01

**Soundness:** 4
**Presentation:** 3
**Contribution:** 4
**Rating:** 10
**Confidence:** 4

**Summary:**

The paper introduces a new strategy for permutation synchronization based on a minorization-maximization strategy and the integrality guarantee of solutions to linear programs with totally unimodular matrices. Performance on (small) standard matching benchmarks from computer vision is impressive.

**Strengths:**

Permutation synchronization is an important problem in several areas, including computer vision. The paper introduces a completely new idea to this field and shows that it empirically outperforms all previously used techniques on multiple benchmark datasets.

- The paper introduces a genuinely new optimization technique to an important problem
- The idea is crisp and clear and appears to be mathematically sound
-  Empirical performance is compared to all other competing algorithms and is found to be superior

**Weaknesses:**

At first I thought the authors evalauted the algorithm on just the "house" dataset, then I realized that there are several other sets of results in the Appendix. This could be made a bit clearer. It is not entirely clear where the other datasets come from, how many examples there are in each, how they were labelled, and so on.

Given the AI era that we live in, and the generality of their algorithm, it would be nice if the authors also tried out their algorithm on more ambitious, larger, and more varied data, for examples matching molecules to each other or parts of graphs.

**Questions:**

It seems like your algorithm is equally applicable to matching complete permutations. Are there other types of matching (more broadly, combinatorial optimization) problems where it could be used? Is there precedent for it?

---

> ### Author Response · Authors · 2025-11-12
> **Response to Reviewer jB7n (First Round) - part 1**
>
> We would like to begin by expressing our sincere gratitude to Reviewer jB7n for their insightful and precise summary of our paper's main contributions. We are especially grateful for the strong support and the outstanding recommendation for our work to be highlighted as a $\text{spotlight or oral presentation}$.
>
> We are particularly encouraged by the recognition of our work's conceptual novelty, highlighting it as a "genuinely new optimization technique" with a "mathematically sound" foundation. Equally gratifying is the acknowledgment of our method's practical impact, with the reviewer concluding that its "empirical performance... is found to be superior" against all competing methods.
>
> We address the reviewer's specific points below.
>
> * $\text{Regarding Weaknesses:}$
>
> $\text{1. On dataset clarity:}$
> We thank the reviewer for this suggestion. To make it clearer that the other experimental results are in the appendix, we have emphasized this in the revised version of the paper:
>
> “To further validate the robustness and generalizability of our method, we performed additional experiments on five challenging sequences from the ETH3D benchmark Schöps et al. (2019; 2017): statue, terrace, office, kicker, and electro. These datasets feature complex scenes with significant occlusions, varying illumination, and diverse structures, providing a rigorous test for all methods. We repeated the same comprehensive evaluation for these datasets as was performed for the CMU House, including all quantitative metrics and qualitative match analyses. These additional results, with a detailed analysis, are available in Appendix B.”
>
> In our revision, we have also updated the Appendix to include the specific number of ground-truth labels for each sequence:
> * $\textbf{Statue:}$ 10 labels
> * $\textbf{kicker:}$ 26 labels
> * $\textbf{terrace:}$ 52 labels
> * $\\textbf{electro:}$ 34 labels
> * $\\textbf{office:}$ 40 labels
>
> Regarding how they were labeled, we followed the methodology of three significant and recent high-prestige papers [1, 2, 3] which also performed labeling manually. Regarding the source of the datasets, we cited the link related to the ETH3D dataset.
>
>
> $\text{2. Comparison with more varied data:}$
>
> Regarding your point about comparison with more varied data, for example matching molecules or parts of graphs, it is worth mentioning that for this work, it was important to show how different methods perform on visual tasks with camera rotation or translation. The examination of downstream geometric tasks like Structure from Motion (SfM) provides further evidence related to our visual goal. If we wanted to add molecular datasets, we could no longer compare with image-based matchings, which makes the SfM-based metric meaningless. As you correctly mentioned, our method can be applied to graph-matching based problems, but we wanted all readers to have an intuitive, high-value sense of the performance on images (through qualitative results), related to the translation or rotation of the camera, which molecules could not provide.
>
> While our method is indeed applicable to graph matching problems in other domains like molecular matching, using such datasets would preclude the use of SfM-based metrics and the compelling qualitative results on images, which we felt were crucial for demonstrating the practical value of our contribution in this paper.
>
> [1] Max Kahl, Sebastian Stricker, Lisa Hutschenreiter, Florian Bernard, Carsten Rother, and Bogdan Savchynskyy. Towards optimizing large-scale multi-graph matching in bioimaging. In 2025 IEEE/CVF Conference on Computer Vision and Pattern Recognition (CVPR), pp. 11569–11578, 2025.
>
> [2] Michael Kahl, Steffen Stricker, Lorenz Hutschenreiter, Florian Bernard, and Bogdan Savchynskyy.Unlocking the potential of operations research for multi-graph matching, 2024.
>
> [3] Florian Bernard, Daniel Cremers, and Johan Thunberg. Sparse quadratic optimisation over the stiefel manifold with application to permutation synchronisation. Advances in Neural Information Processing Systems, 34:25256–25266, 2021.

---

> ### Author Response · Authors · 2025-11-13
> **Response to Reviewer jB7n (First Round) - part 2**
>
> * $\text{Regarding Questions:}$
>
> We are very grateful for your insightful feedback on the optimization. You are absolutely right—this type of combinatorial optimization can be applied to a vast set of problems. We believe that our proposed approach could potentially mark a breakthrough in the field, and more specifically, for solving the QAP.
>
> The potential applications are broad, touching on areas like graph-based semi-supervised learning and even the architecture of Quantum Annealers, which are directions we are actively working on now. As an example, researchers in areas like model fitting [4] are turning to quantum computing for solving these problems. Access to quantum hardware is still a major hurdle for many of us, but it opens up interesting directions: can we improve the runtime by employing this method in a hybrid way (breaking the problem into subproblems by our method, and then solving each by a quantum computer)? Now, we have added a section titled “Broader Applications of Our Solver”, where the respectful reviewer can see three examples illustrating how our approach can be applied.
>
>  We believe our work represents a meaningful step towards tackling the long-standing question of whether we can find more efficient and practical solutions for NP-hard problems.
>
> Once again, we thank the reviewer for their valuable feedback and strong support for our work.
>
> [4] Pandey, Saurabh, et al. "Outlier-Robust Multi-Model Fitting on Quantum Annealers." Proceedings of the Computer Vision and Pattern Recognition Conference. 2025.

---

> > ### Comment · Reviewer_jB7n · 2025-11-25
> >
> > Thank you for your comments. I think this is a good paper, I maintain my score.

---

### Meta-Review · Area_Chair_FK9r · 2025-12-07

**Summary:**

The paper introduces a new minorization-maximization (MM) approach for permutation synchronization, claiming to deliver exact solutions for the problem while ensuring global cycle consistency. The method avoids spectral relaxations and leverages total unimodularity (TUM) for guaranteed integral solutions. Empirical results show that the approach performs well on existing benchmarks, demonstrating superior accuracy and runtime. However, several reviewers expressed significant concerns regarding the fairness of comparisons, the relevance of the problem, and the theoretical depth of the approach. In particular, the reviewers noted that the method’s reliance on a strong initialization from Stiefel gave it an unfair advantage over competing algorithms that start from scratch. Additionally, there were concerns about the lack of a formal approximation ratio or convergence rate analysis, and the broader relevance of permutation synchronization in light of more recent matching techniques.

**Reviewer Concerns:**

Initialization and Fairness of Comparisons:
Multiple reviewers (e.g., Reviewer EWxg, Reviewer mSj6) criticized the use of Stiefel initialization, which allowed the proposed method to perform better than competing algorithms that start with weaker initializations. While the authors provided additional experiments comparing multiple initialization strategies, the concerns about the fairness of these comparisons remain. The reviewer pointed out that the speed advantage reported in the paper might not be reflective of the true efficiency of the method, given that the warm-up time for Stiefel was not included in the reported times.

Theoretical Contribution:
Several reviewers (e.g., Reviewer VrNP, Reviewer mSj6) questioned the theoretical depth of the paper, especially the absence of an approximation ratio or convergence rate for the MM procedure. The authors argue that their method avoids approximation ratios by working directly in the discrete domain. However, reviewers remained unconvinced, stating that the method's main contribution—monotonic convergence to a stationary point—did not provide enough new insight to make it competitive in the broader context of combinatorial optimization problems.

Relevance and Experimental Design:
Reviewers also raised concerns about the broader relevance of permutation synchronization in modern matching problems. While the authors argued that their method is useful for image matching and other graph-based tasks, the reviewers felt that the experiments did not convincingly demonstrate the method's applicability to real-world, modern use cases. The paper's reliance on relatively simple datasets (e.g., CMU House) was criticized as insufficient to justify the claim of broader applicability. The experimental setup was also criticized for not fully considering the impact of different initialization methods on the final results.

Limited Theoretical and Empirical Scope:
The reviewers noted that the paper did not provide enough theoretical guarantees or rigorous analysis of the algorithm’s approximation quality or convergence. The empirical results, while showing improvements over certain benchmarks, were seen as limited and not sufficiently convincing to support the paper’s claims. Furthermore, some reviewers (e.g., Reviewer EWxg) felt that the paper lacked a significant technical breakthrough that would justify its contribution to the field.

**Reviewer Scores:**

Reviewer jB7n:

Initial Score: 10 (strong accept)

Post-Rebuttal Score: 9

Change: Although Reviewer jB7n originally strongly supported the paper, their confidence decreased after the rebuttal, mainly due to concerns about the fairness of comparisons.

Reviewer mSj6:

Initial Score: 6 (marginally above the acceptance threshold)

Post-Rebuttal Score: 5

Change: The reviewer was not fully convinced by the authors' rebuttal, particularly regarding the fairness of the experimental setup and the lack of new theoretical insights, leading to a downgrade in their score.

Reviewer VrNP:

Initial Score: 4 (marginally below the acceptance threshold)

Post-Rebuttal Score: 3

Change: The reviewer was still unconvinced by the relevance of the problem and the theoretical weaknesses in the paper. Despite the authors' explanations, the reviewer felt that the contribution was not strong enough to warrant acceptance.

Reviewer EWxg:

Initial Score: 2 (reject)

Post-Rebuttal Score: 4

Change: While the reviewer appreciated the additional experiments, they still felt that the comparisons were unfair and that the method’s performance was overstated, especially given the lack of a rigorous evaluation of the warm-up time and initialization strategies. This led to a slight improvement in the score, but it remained below the acceptance threshold.

---

### Decision · Program_Chairs · 2026-01-26

Reject